# Neuropilin-1 functions as a VEGFR2 co-receptor to guide developmental angiogenesis independent of ligand binding

Maria V Gelfand[1†], Nellwyn Hagan[1†], Aleksandra Tata[1†], Won-Jong Oh[1,2], Baptiste Lacoste[1], Kyu-Tae Kang[3,4,7], Justyna Kopycinska[1], Joyce Bischoff[3,4], Jia-Huai Wang[5,6], Chenghua Gu[1]*

[1]Department of Neurobiology, Harvard Medical School, Boston, United States; [2]Korea Brain Research Institute, Daegu, Republic of Korea; [3]Vascular Biology Program, Boston Children's Hospital, Harvard Medical School, Boston, United States; [4]Department of Surgery, Boston Children's Hospital, Harvard Medical School, Boston, United States; [5]Department of Medical Oncology, Dana-Farber Cancer Institute, Harvard Medical School, Boston, United States; [6]Department of Cancer Biology, Dana-Farber Cancer Institute, Harvard Medical School, Boston, United States; [7]College of Pharmacy, Duksung Women's University, Seoul, Republic of Korea

**Abstract** During development, tissue repair, and tumor growth, most blood vessel networks are generated through angiogenesis. Vascular endothelial growth factor (VEGF) is a key regulator of this process and currently both VEGF and its receptors, VEGFR1, VEGFR2, and Neuropilin1 (NRP1), are targeted in therapeutic strategies for vascular disease and cancer. NRP1 is essential for vascular morphogenesis, but how NRP1 functions to guide vascular development has not been completely elucidated. In this study, we generated a mouse line harboring a point mutation in the endogenous *Nrp1* locus that selectively abolishes VEGF-NRP1 binding (*Nrp1^VEGF−^*). *Nrp1^VEGF−^* mutants survive to adulthood with normal vasculature revealing that NRP1 functions independent of VEGF-NRP1 binding during developmental angiogenesis. Moreover, we found that *Nrp1*-deficient vessels have reduced VEGFR2 surface expression in vivo demonstrating that NRP1 regulates its co-receptor, VEGFR2. Given the resources invested in NRP1-targeted anti-angiogenesis therapies, our results will be integral for developing strategies to re-build vasculature in disease.

*For correspondence: Chenghua_Gu@hms.harvard.edu

†These authors contributed equally to this work

Competing interests: The authors declare that no competing interests exist.

## Introduction

Blood vessels provide oxygen and nutrients to cells throughout the body and are essential for tissue homeostasis and repair as well as tumor growth. The molecular mechanisms underlying angiogenesis have become increasingly clear, and VEGF is an essential player in this process (*Carmeliet et al., 1996, 1999; Ferrara et al., 1996, 2003; Iruela-Arispe and Dvorak, 1997; Miquerol et al., 1999; Ruhrberg et al., 2002; Stalmans et al., 2002; Rossant and Hirashima, 2003; Maes et al., 2004; Coultas et al., 2005; Olsson et al., 2006; Chung and Ferrara, 2011*). VEGF operates by interacting with three receptors, VEGFR1, VEGFR2 (KDR/Flk1), and NRP1 (*Ferrara et al., 2003; Chung and Ferrara, 2011*). Although these three receptors are expressed in spatially and temporally overlapping patterns, they are thought to play different roles in VEGF signaling. The main receptor for VEGF, VEGFR2, is a receptor tyrosine kinase whose activity is crucial for VEGF signaling (*Olsson et al., 2006*). Upon binding VEGF, VEGFR2 phosphorylates intracellular targets leading to a multitude of cellular responses including proliferation,

**eLife digest** Blood flows through blood vessels to carry oxygen and nutrients towards, and waste away from, the cells of the body. New blood vessels are formed not only during development but also throughout life as part of normal tissue growth and repair. However, blood vessels may also form as a consequence of diseases, such as cancer. For example, tumors often stimulate the growth of new blood vessels to ensure a good supply of blood carrying nutrients and oxygen. As such, some anti-cancer therapies try to stop blood vessels from developing in an attempt to slow down or prevent tumor growth.

New blood vessels often form by branching off from existing vessels. One molecule that stimulates this branching process is called vascular endothelial growth factor (or VEGF for short). Three 'receptor' proteins found on the outside of cells can bind to the VEGF molecule and then trigger a response inside the cell that guides the development of new blood vessels. VEGF and its receptor proteins—including one called NRP1—are being investigated as a possible target for drugs that could treat cancer and other diseases affecting blood vessels. However, the exact mechanisms that control the formation of new blood vessels are not fully understood, which makes it difficult to develop these treatments.

Now Gelfand et al. have created mice whose NRP1 receptors cannot bind VEGF. These mice unexpectedly survive to adulthood and develop normal blood vessels. This outcome is in contrast to mice that lack NRP1, which normally die as embryos and have severe defects with their nerves and blood vessels. Gelfand et al. instead found that mice that only lack NRP1 in the cells of their blood vessels had less of another receptor protein called VEGFR2 on the surface of these cells. This result suggests that NRP1 controls blood vessel development, not by binding to VEGF but by affecting how much of the VEGFR2 receptor is available to interact with VEGF.

These findings challenge the long-held view of how NRP1 functions and lead Gelfand et al. to suggest a new mechanism: NRP1 interacts with VEGFR2, rather than with VEGF, to control the formation of new blood vessels. Future work will aim to uncover how these interactions regulate the normal development of blood vessels, and if other molecules that bind to NRP1 are involved in this process. Furthermore, these findings may help to guide the on-going efforts to develop drugs that target NRP1 into treatments that are effective against diseases that involve problems with blood vessels—including diabetes, immune disorders, and cancer.

migration, and transcriptional modification via signaling pathways such as PI3K, Src, and PLCY (*Olsson et al., 2006*). In contrast, NRP1 is a multifaceted transmembrane receptor that not only binds VEGF and forms a complex with VEGFR2 but also binds a structurally and functionally unrelated family of traditional axon guidance cues, the secreted class 3 semaphorins (SEMA3) (*He and Tessier-Lavigne, 1997*; *Kolodkin et al., 1997*; *Soker et al., 1998*). Consistent with these binding partners, $Nrp1^{-/-}$ mice are embryonically lethal with both neural and vascular defects (*Kitsukawa et al., 1997*; *Kawasaki et al., 1999*), indicating that NRP1 protein is instrumental for developmental angiogenesis. However, how NRP1 functions in conjunction with multiple ligands and receptors to guide vascular development remains elusive.

Previous work has started to systematically dissect NRP1 function in vivo using a combination of structure–function analyses and mouse genetic approaches. In particular, endothelial-specific NRP1 knock-outs ($Tie2$-$Cre;Nrp1^{fl/-}$) recapitulate the devastating vascular defects observed in $Nrp1^{-/-}$ mice— the vascular network is poorly developed and large endothelial cell aggregates form within the brain (*Gu et al., 2003*). This result strongly demonstrates that NRP1 is a cell autonomously required in endothelial cells for its absolutely essential function in developmental angiogenesis. To pinpoint how SEMA3-NRP1 vs VEGF-NRP1 binding contributes to NRP1's critical role in vascular development, previous work generated a knock-in mouse line, $Nrp1^{Sema-}$, in which SEMA3-NRP1 interactions were abolished and VEGF-NRP1 binding was maintained (*Gu et al., 2003*). $Nrp1^{Sema-}$ mice mimicked the neural defects observed in the $Nrp1^{-/-}$ but did not exhibit any vascular abnormalities. These data suggest that SEMA3-NRP1 binding does not mediate NRP1's important function in vascular morphogenesis and instead point to the hypothesis that VEGF-NRP1 interactions may be integral for angiogenesis.

Currently, the dominant view in the field asserts that VEGF-NRP1 binding enhances VEGFR2 activity and downstream signaling. Yet, the functional consequence of VEGF-NRP1 interactions has only been studied indirectly using in vitro methodology and blocking antibodies in vivo (*Pan et al., 2007*; *Herzog et al., 2011*). Specifically, an antibody inhibiting VEGF-NRP1 binding was found to interfere with retinal vascular remodeling as well as tumor angiogenesis (*Pan et al., 2007*) and is currently being developed as a therapeutic strategy to block vessel outgrowth. This study suggests that VEGF-NRP1 binding facilitates pathological angiogenesis. However, in vivo evidence describing a role for VEGF-NRP1 binding in vascular development is currently lacking and the precise function of NRP1 in VEGF-mediated angiogenesis urgently needs to be addressed.

To delineate the role of VEGF-NRP1 interactions, we identified a single amino acid residue in the b1 domain of NRP1 that is necessary for VEGF-NRP1 binding and generated a mouse harboring this point mutation to abolish VEGF-NRP1 interactions in vivo ($Nrp1^{VEGF-}$). Surprisingly, although VEGF-NRP1 binding was successfully eliminated, the $Nrp1^{VEGF-}$ mutants survived into adulthood and did not display any of the severe vascular phenotypes seen in either the $Nrp1^{-/-}$ or the endothelial-specific NRP1 knock-out. Upon closer examination, NRP1-deficient blood vessels in the endothelial-specific NRP1 knock-out exhibited reduced VEGFR2 surface expression, a phenomenon not observed in the $Nrp1^{VEGF-}$ mutant. These results challenge the well-accepted view that NRP1 requires VEGF-NRP1 binding to facilitate developmental angiogenesis and points to a provocative new hypothesis that the angiogenic role of NRP1 lies in its capacity as a VEGFR2 co-receptor. Interestingly, retinal angiogenesis and blood flow recovery following hindlimb ischemia were mildly perturbed in the $Nrp1^{VEGF-}$ mutant suggesting that the postnatal vascular system is uniquely sensitive to the loss of VEGF-NRP1 binding. Together, this work not only significantly advances our basic scientific understanding of how NRP1 functions in VEGF-mediated angiogenesis, but also provides new insights that may facilitate the development of more effective NRP1-targeted anti-angiogenesis therapies.

## Results

### Identification of an *Nrp1* mutation that abolishes VEGF-NRP1 binding

We sought to elucidate the in vivo function of VEGF-NRP1 binding by generating a mouse line that selectively disrupts VEGF binding to NRP1. A previous structure–function analysis revealed that the b1 domain of NRP1 is necessary and sufficient for VEGF binding (*Gu et al., 2002*). However, this b1 region is also required for SEMA3-NRP1 interactions, so a series of *Nrp1* variants containing smaller deletions in the b1 domain were engineered with site-directed mutagenesis to identify a region specific for VEGF-NRP1 binding (*Figure 1A*). Based upon previous publications, we first targeted two specific sites in the b1 domain: the 7-residue binding site of the Pathologische Anatomie Leiden-Endothelium (PAL-E) monoclonal antibody which competes with VEGF for NRP1 binding (*Jaalouk et al., 2007*) and the 3-residue binding site of the VEGF analog tuftsin (*Vander Kooi et al., 2007*) (*Figure 1A–B*). COS-1 cells were transfected with wild-type (WT) or mutant *Nrp1* constructs and assessed for NRP1 expression. PAL-E and tuftsin binding site mutations did not affect NRP1 protein expression at the cell surface as examined by non-permeabilized antibody staining (*Figure 1C*, *Figure 1—figure supplement 1*). Ligand binding to NRP1 was assessed using alkaline phosphatase-tagged VEGF (AP-VEGF) and SEMA3A (AP-SEMA3A) in conjunction with alkaline phosphatase histochemistry. All of the PAL-E or tuftsin binding site variants were capable of abolishing VEGF-NRP1 binding, but unfortunately, also eliminated SEMA3-NRP1 binding (*Figure 1C*, *Figure 1—figure supplement 1*).

We decided to use an unbiased approach and designed our subsequent *Nrp1* variants based upon the crystal structure of the full NRP1 b1 domain. Specifically, we identified a hydrophilic region comprised of several negatively charged residues that provided a promising mutagenesis site for abolishing VEGF-NRP1 binding (*Figure 1A*). Several of these residues were mutated to amino acids of the opposite charge in order to preserve the hydrophilic nature of the region. As with previous *Nrp1* variants, NRP1 surface expression was unperturbed in transfected COS-1 cells (*Figure 1C*). One of these mutations (E282K) did not affect the binding of either AP-SEMA3A or AP-VEGF, while others (E282K and E420K) eradicated binding of both the ligands (*Figure 1—figure supplement 1*). However, the D320K mutation converting aspartic acid 320 into lysine ($Nrp1^{D320K}$) successfully abolished VEGF-NRP1 binding while conserving AP-SEMA3A binding as demonstrated through alkaline phosphatase histochemical staining on transfected COS-1 cells (*Figure 1C*, *Figure 2A,C*). Moreover, the $Nrp1^{D320K}$ mutation also abolished the binding of other VEGF family members including Placenta Growth Factor

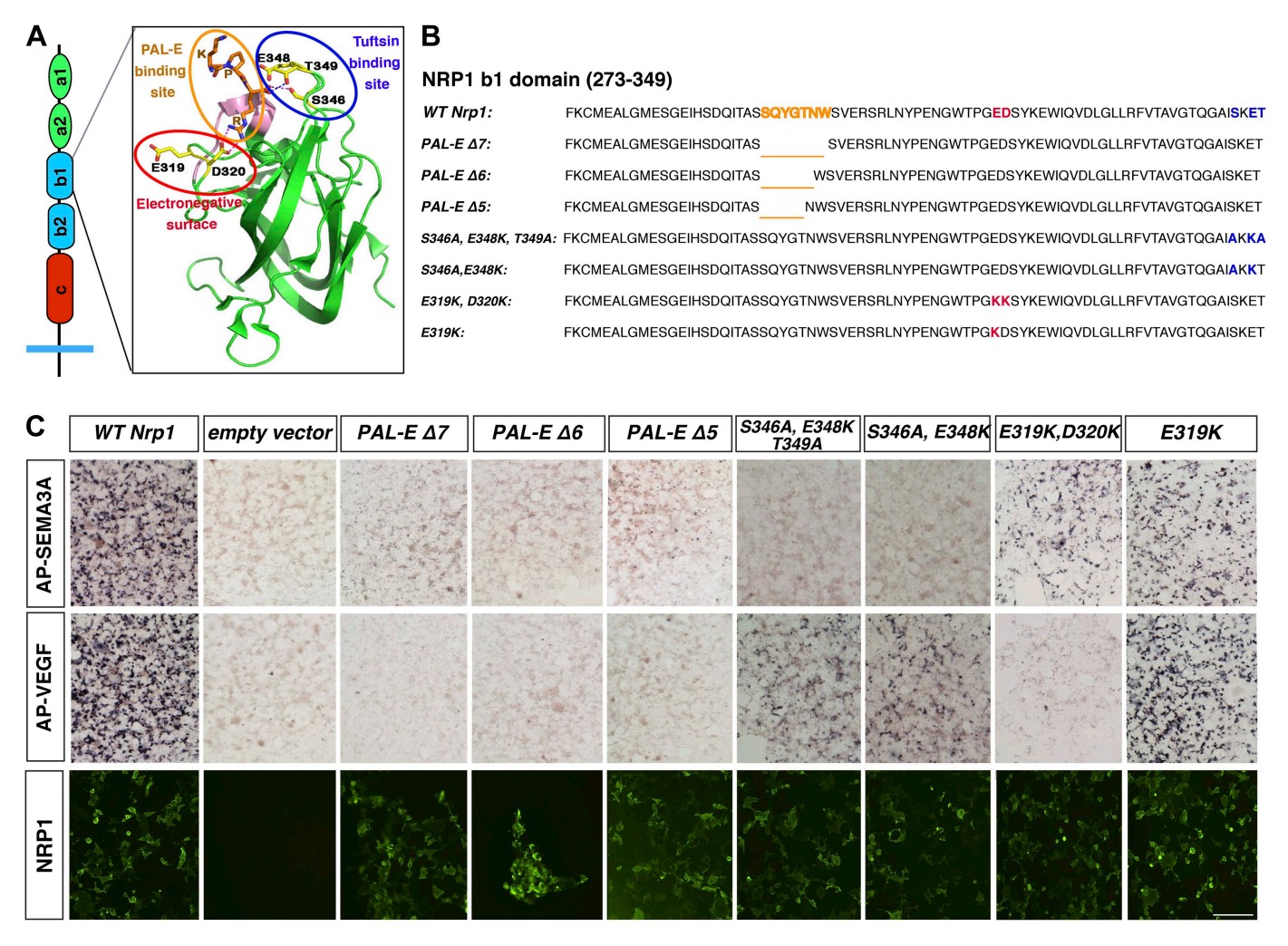

**Figure 1**. Design and assessment of *Nrp1* variants harboring mutations in the VEGF-binding site. (**A**) Schematic representation of the NRP1 b1 extracellular domain and crystal structure highlighting three potential mutagenesis sites: the PAL-E binding site (orange circle), tuftsin binding site (blue circle), and electronegative surface (red circle). (**B**) Sequence of the *Nrp1* b1 domain indicating the deletion or mutation sites for the candidate constructs. (**C**) AP-SEMA3A (top row) or AP-VEGF (middle row) binding to COS-1 cells overexpressing the indicated constructs. Deletion of the entire PAL-E binding site (*Nrp1^PAL-EΔ7^*) or partial deletion of the PAL-E binding site (*Nrp1^PAL-EΔ6^* and *Nrp1^PAL-E Δ5^*) eliminated both AP-SEMA3A and AP-VEGF binding. Likewise, mutations in the tuftsin binding site (S346A, E348A, T349A or S346A, E348A) abolished AP-SEMA3A binding and reduced AP-VEGF binding. Although mutations in the NRP1 electronegative surface (E319K, D320K) eliminated AP-VEGF binding and reduced AP-SEMA3A binding, the E319K mutation only slightly reduced AP-SEMA3A binding and maintained AP-VEGF binding. Antibody staining of unpermeabilized cells (lower row) demonstrated normal NRP1 surface expression. Scale bar: 50 μm.

The following figure supplement is available for figure 1:

**Figure supplement 1**. Assessment of additional *Nrp1* variants containing mutations in the VEGF-binding site.

(PlGF) and Vascular Endothelial Growth Factor B (VEGFB) (***Figure 2—figure supplement 1***). In a liquid alkaline phosphatase activity assay, *Nrp1^D320K^* was co-expressed with *PlexinA4* (*Plex4A*) to more accurately reflect the in vivo situation in which SEMA3A signals through a holoreceptor complex of both NRP1 and PlexinA. AP-SEMA3A binding levels to WT NRP1 and NRP1^D320K^ were indistinguishable (***Figure 2D***), and the dissociation constant (K_D) of SEMA3A-NRP1^D320K^/PlexinA4 was unchanged from that of SEMA3A-NRP1/PlexinA4 further verifying that the SEMA3A-NRP1/PlexinA4 interaction was intact (***Figure 2E***). Finally, Western blot analysis confirmed that NRP1 protein expression levels were equivalent in COS-1 cells transfected with *WT Nrp1* and *Nrp1^D320K^* (***Figure 2B***). Taken together, these

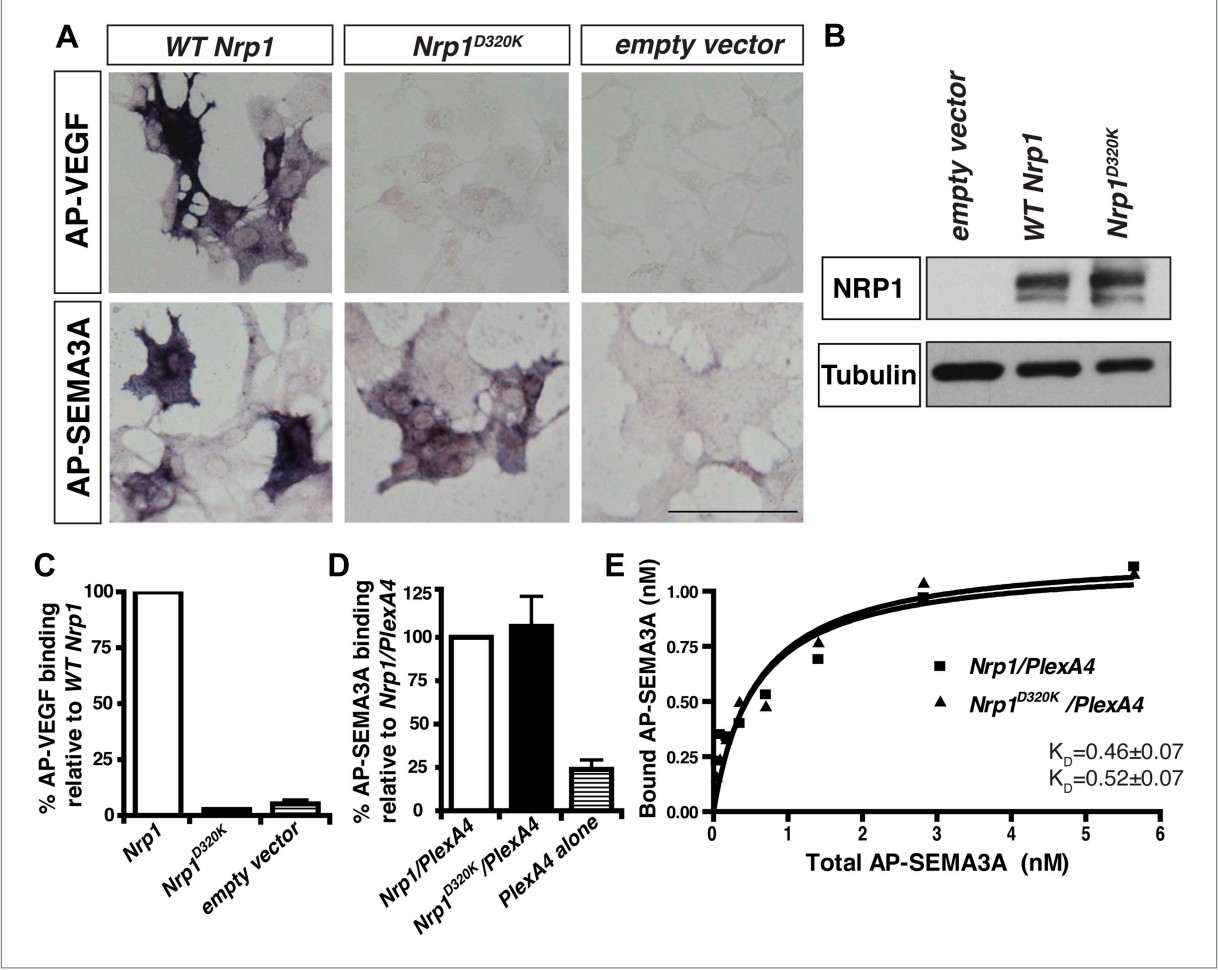

**Figure 2**. The Nrp1[D320K] mutation selectively eliminates VEGF-NRP1 binding in vitro. (**A**) AP-VEGF binding in COS-1 cells overexpressing the indicated *Nrp1* construct. WT NRP1 bound AP-VEGF strongly, while AP-VEGF binding to NRP1[D320K] was abolished. Scale bar: 100 μm (**B**) Western blot shows that equivalent levels of NRP1 protein in COS-1 cells transfected with the *WT Nrp1* and *Nrp1[D320K]*. (**C**) Quantification of the binding assay shows that AP-VEGF-NRP1[D320K] binding was abolished even after normalization for protein content and NRP1 expression. (**D**) Quantification of AP-SEMA3A binding shows comparable AP-SEMA3A binding to WT NRP1 and NRP1[D320K]. (**E**) Measurement of the dissociation constant ($K_D$) of AP-SEMA3A demonstrates that AP-SEMA3A bound to the NRP1[D320K]/PlexA4 complex with the same affinity as the NRP1/PlexA4 complex.

The following figure supplement is available for figure 2:

**Figure supplement 1**. VEGFA, VEGFB, and PLFG binding to NRP1 was abolished in the *Nrp1[D320K]* mutant.

data demonstrate that the *Nrp1[D320K]* mutation is sufficient to eliminate VEGF binding and maintain SEMA3A binding in vitro.

## Generation and validation of the *Nrp1[VEGF−]* mouse line

A gene replacement strategy was implemented to generate a mouse line harboring the *Nrp1[D320K]* mutation in the endogenous *Nrp1* locus, delineated as *Nrp1[VEGF−]*. Specifically, two base pair mutations were introduced into exon 6 of the mouse *Nrp1* gene to produce the D320K mutation in the endogenous Asp320 location (**Figure 3A**). After recombineering, embryonic stem cells were screened via PCR and sequenced to confirm that the D320K mutation was appropriately introduced into the *Nrp1* locus (**Figure 3—figure supplement 1A–C**). Once *Nrp1[VEGF−]* mice were obtained, the presence of the D320K mutation was verified by sequencing (**Figure 3—figure supplement 1D**). Importantly, the *Nrp1[VEGF−]* mutants expressed normal levels of NRP1 protein as assessed by Western blot on embryonic day 14.5 (E14.5) lung and adult heart, brain, lung and kidney (**Figure 3C**, **Figure 3—figure supplement 2D**). AP-VEGF and AP-SEMA3A binding was examined at E12.5 in the dorsal root entry zone (DREZ),

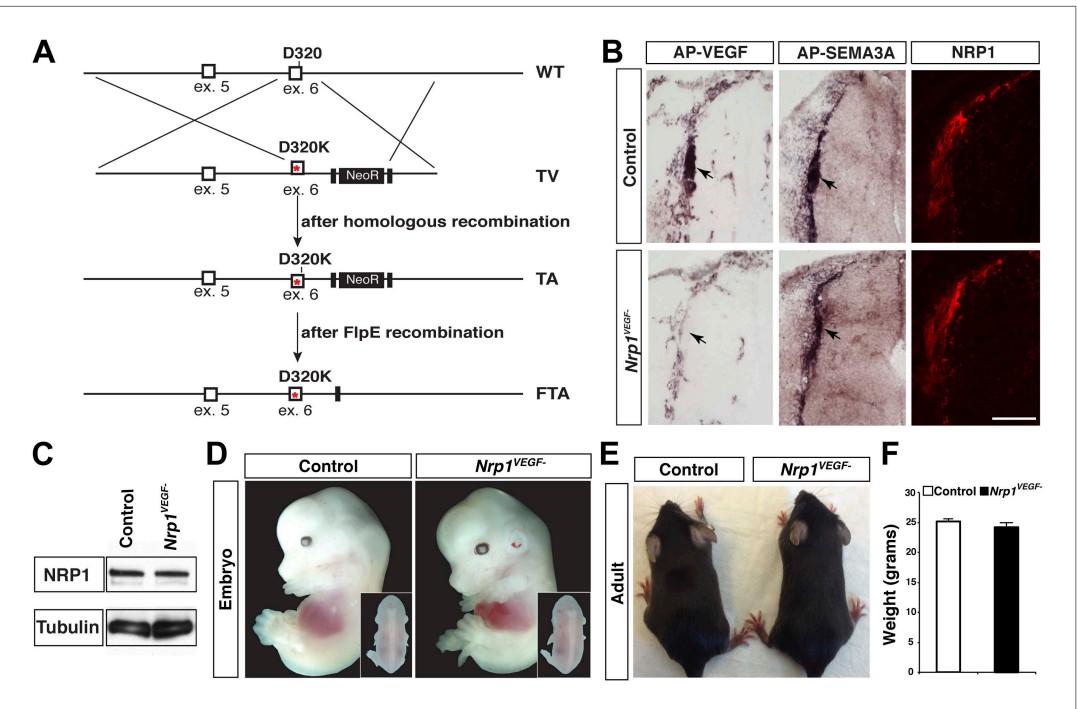

**Figure 3**. *Nrp1^{VEGF-}* mice selectively abolish VEGF-NRP1 binding in vivo. (**A**) Targeting vector design for the generation of *Nrp1^{VEGF−}* mice. The WT genomic region contained residue D320 in exon 6 of *Nrp1*. The targeting vector (TV) introduced the D320K mutation along with an Frt-flanked NeoR cassette to form the targeted allele (TA). After FlpE-mediated excision of the NeoR cassette, the final targeted allele (FTA) had the D320K mutation as well as one remaining Frt site. (**B**) Section binding assays demonstrated that AP-VEGF binding to the dorsal root entry zone (DREZ) was abolished in the *Nrp1^{VEGF−}* mutants (arrows, left panels) while AP-SEMA3A binding to the DREZ appeared similar between *Nrp1^{VEGF−}* and control animals (arrows, middle panels). Scale bar: 100 µm. (**C**) Western blot from E14.5 lung tissue shows that NRP1 protein level was not affected in *Nrp1^{VEGF−}* animals. (**D** and **E**) *Nrp1^{VEGF−}* mutants appear indistinguishable from controls littermates at embryonic (E14.5) and adult stages. (**F**) *Nrp1^{VEGF−}* mutants exhibit normal body weight in adulthood (n = 7, males).

The following figure supplements are available for figure 3:

**Figure supplement 1**. Screening and verification of ES cells for the generation of the *Nrp1^{VEGF−}* mutant.

**Figure supplement 2**. The *Nrp1^{VEGF−}* mutant mice exhibit normal gross morphology.

where NRP1-expressing axons from the dorsal root ganglion enter the spinal cord. Both AP-VEGF and AP-SEMA3A bound to the DREZ in control animals (*Figure 3B*) while AP-VEGF binding to the DREZ was abolished in the *Nrp1^{VEGF−}* mutant (*Figure 3B*), confirming that this mutation eliminated VEGF-NRP1 binding in vivo. Moreover, NRP1 immunostaining and AP-SEMA3A binding to the DREZ appeared similar between *Nrp1^{VEGF−}* and control littermates (*Figure 3B*). Finally, the *Nrp1^{VEGF−}* mutants failed to display the perinatal lethality or cardiac defect observed in the *Nrp1^{Sema−}* mutants (*Gu et al., 2003*), further confirming functional SEMA3-NRP1 binding in *Nrp1^{VEGF−}* mice (*Figure 3— figure supplement 1*).

## VEGF-NRP1 binding is not required for developmental angiogenesis

Despite the embryonic lethality previously described in *Nrp1^{−/−}* and *Tie2-Cre,Nrp1^{fl/−}* animals, *Nrp1^{VEGF−}* mice were born at expected Mendelian ratios and maintained their vitality into adulthood (p > 0.05 for observed vs expected, *Figure 3—figure supplement 2E*). The *Nrp1^{VEGF−}* mutants exhibited normal gross morphology throughout embryonic and postnatal stages (*Figure 3D,E*) and failed to develop the cardiac defects previously observed in the *Nrp1^{−/−}*, *Tie2-Cre;Nrp1^{fl/−}*, and *Nrp1^{Sema−}* mutants (*Figure 3— figure supplement 2A*). Moreover, *Nrp1^{VEGF−}* animals displayed normal body weight (*Figure 3F*), organ growth (*Figure 3—figure supplement 2B,C*), and fertility.

To thoroughly examine vascular integrity during development, isolectin staining was employed to visualize blood vessels in embryonic and perinatal brain sections and vessel ingression, morphology, and branching were assessed in the *Nrp1*$^{VEGF-}$ mutant. Surprisingly, *Nrp1*$^{VEGF-}$ animals did not exhibit any of the vascular abnormalities observed in the endothelial-specific NRP1 knock-out. As shown in *Figure 4A* and quantified in *Figure 4B–C*, cortical vessel ingression was nearly absent in *Tie2-Cre;Nrp1*$^{fl/fl}$ animals at E11.5 while ingression was unaffected in the *Nrp1*$^{VEGF-}$ mutants. In addition, *Tie2-Cre;Nrp1*$^{fl/fl}$ animals had abnormally large vascular aggregates distributed throughout the striatum at E14.5 while vessels were evenly dispersed without aggregates in both control and *Nrp1*$^{VEGF-}$ animals (*Figure 4D–F*). Finally, *Tie2-Cre;Nrp1*$^{fl/fl}$ animals had a significant decrease in vessel branching in the cortex at E14.5 while *Nrp1*$^{VEGF-}$ animals exhibited normal vessel branching (*Figure 4G–I*). Moreover, unlike the endothelial-specific NRP1 knock-out, the long term viability of the *Nrp1*$^{VEGF-}$ mutants allowed us to assess cortical vessel branching and coverage at P7 which was indistinguishable from control littermates (*Figure 4G–I*, *Figure 4—figure supplement 1*). Therefore, VEGF-NRP1 binding is not required for developmental angiogenesis.

## NRP1 functions to modulate VEGFR2 levels independent of VEGF-NRP1 binding

The normal developmental angiogenesis observed in the *Nrp1*$^{VEGF-}$ mutants clearly demonstrates that VEGF-NRP1 binding is not responsible for the vascular defects observed in *Nrp1*$^{-/-}$ or endothelial-specific NRP1 knock-outs. In this regard, NRP1 must function through an alternative mechanism to regulate vascular development during embryogenesis. The intracellular domain of NRP1 does not have any obvious enzymatic activity and is not responsible for the signal transduction mediating angiogenesis (*Fantin et al., 2011*; *Lanahan et al., 2013*). Therefore, two apparent alternatives remain. One possibility is that a yet unidentified ligand outside the VEGF or SEMA3 family binds to NRP1 and instructs developmental angiogenesis. Alternatively, NRP1 may control vascular development by directly regulating its co-receptor, VEGFR2.

To test this second possibility, VEGFR2 expression was evaluated in the *Tie2-Cre;Nrp1*$^{fl/-}$ mutants and control littermates via Western blot on E14.5 lung tissue. This biochemical assay revealed that total VEGFR2 protein levels were significantly reduced in the *Tie2-Cre;Nrp1*$^{fl/-}$ mutants compared to their control littermates (*Figure 5A–B*). To determine the cell surface expression of VEGFR2 in vivo, we used fluorescence-activated cell sorting (FACS) to specifically quantify VEGFR2 expression at the cell surface of non-permeabilized endothelial cells derived from the acutely dissociated lungs of *Tie2-Cre;Nrp1*$^{fl/-}$ and control embryos. Remarkably, *Tie2-Cre;Nrp1*$^{fl/-}$ mutants displayed a significant decrease in the fluorescence intensity of VEGFR2 labeling as compared to control littermates (*Figure 5E–F*), suggesting that NRP1 functions to regulate VEGFR2 surface expression in endothelial cells. In contrast, both Western blot and FACS analysis determined that VEGFR2 protein levels were unperturbed in *Nrp1*$^{VEGF-}$ animals (*Figure 5C–D,G–F*). In addition, co-immunoprecipitation on P7 lung tissue revealed that NRP1 and VEGFR2 are physically associated in both control and *Nrp1*$^{VEGF-}$ animals (*Figure 5—figure supplement 1B*), validating that NRP1-VEGFR2 receptor complex formation does not require VEGF-NRP1 binding in vivo. This result mimics our co-immunoprecipitation experiments on HEK293T cells transfected with either *WT Nrp1* or *Nrp1*$^{D320K}$ constructs (*Figure 5—figure supplement 1A*). Together, these findings indicate that NRP1 plays a role in regulating the cell surface expression of VEGFR2 in endothelial cells and that VEGF-NRP1 binding is not necessary for this function in vivo (*Figure 5G*).

To examine VEGF signaling in the *Tie2-Cre;Nrp1*$^{fl/-}$ and *Nrp1*$^{VEGF-}$ mutants, VEGFR2 phosphorylation was examined via Western blot on embryonic lung tissue isolated at E14.5. Specifically, *Tie2-Cre;Nrp1*$^{fl/-}$ mutants had a severe reduction in VEGFR2 phosphorylation at the tyrosine residue 1175 (Y1175) upon VEGF treatment (*Figure 5—figure supplement 2A,B*). Interestingly, *Nrp1*$^{VEGF-}$ mutants also exhibited a mild reduction in VEGFR2 phosphorylation while total VEGFR2 protein levels were well maintained (*Figure 5—figure supplement 2C,D*). Although the level of pVEGFR2 in the *Nrp1*$^{VEGF-}$ mutant was sufficiently high to support vascular development during embryogenesis, the modest reduction in pVEGFR2 may manifest in issues with angiogenesis, vascular maintenance, and regeneration in the postnatal animal.

## VEGF-NRP1 binding is required for postnatal angiogenesis

To directly test the role for VEGF-NRP1 binding in postnatal angiogenesis, whole-mount staining was performed with isolectin and an antibody against α-smooth muscle actin (α-SMA) to visualize the retinal blood vessels and arteries, respectively. At P9, the *Nrp1*$^{VEGF-}$ mutants exhibited a reduction in

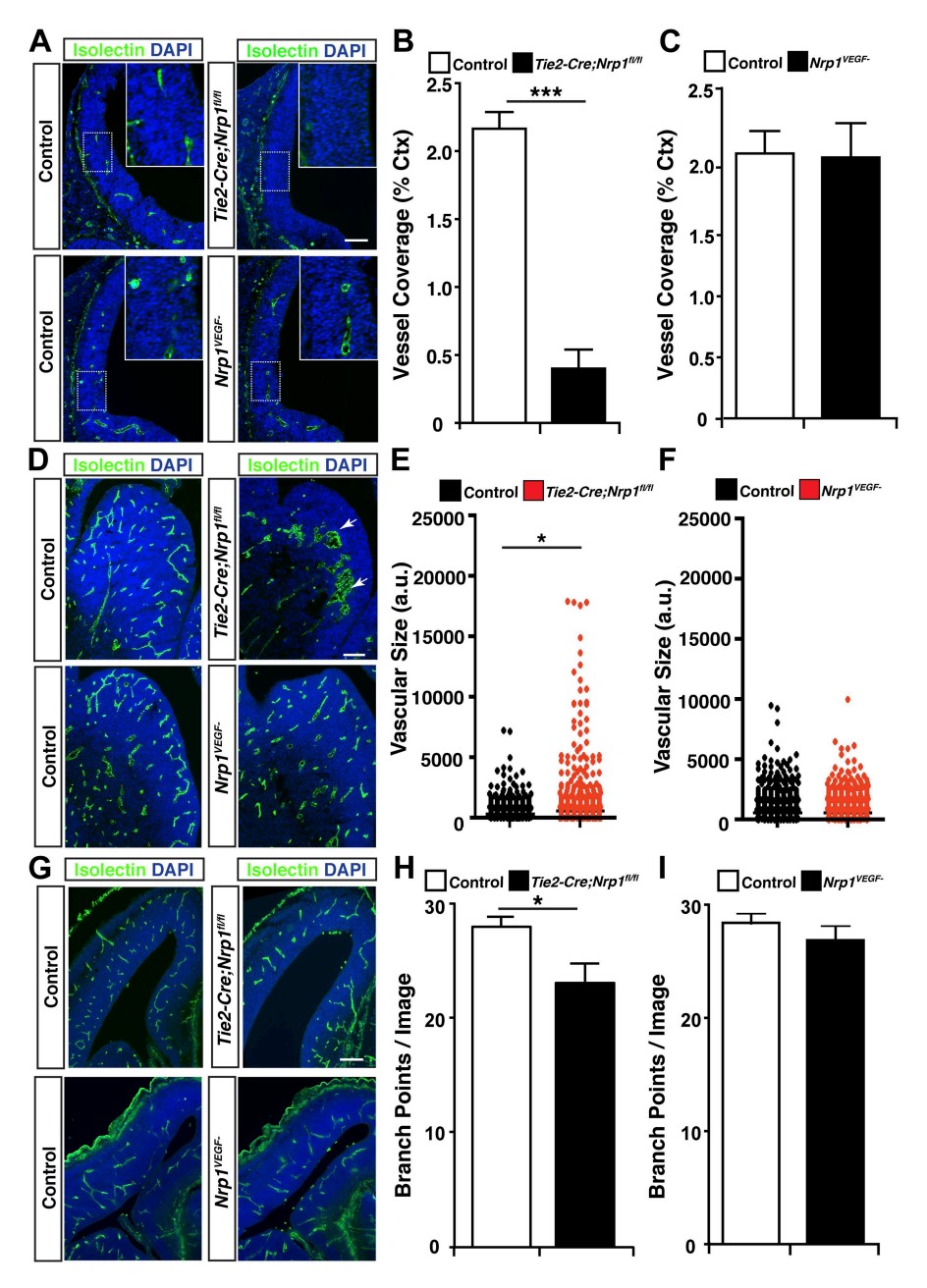

**Figure 4**. VEGF-NRP1 binding is not required for developmental angiogenesis. (**A**) Vessel staining with isolectin (green) revealed that *Tie2-Cre;Nrp1^fl/fl^* mutants had delayed vessel ingression into the cerebral cortex at E11.5 while the *Nrp1^VEGF−^* mutants exhibited normal ingression. DAPI was used to stain the nuclei (blue). (**B** and **C**) Quantification of cortical vessel ingression shown in **A**, n = 3. (**D**) *Tie2-Cre;Nrp1^fl/fl^* mutants exhibited large vessel clumps in the brain (particularly in the striatum) at E14.5, a phenotype not observed in the *Nrp1^VEGF−^* mutants. (**E** and **F**) Quantification of vessel size in E14.5 striatum shown in **D**, n = 3. (**G**) *Tie2-Cre;Nrp1^fl/fl^* mutants had reduced vessel branching in the cerebral cortex while the *Nrp1^VEGF−^* mutants displayed normal vessel branching at E14.5. (**H** and **I**) Quantification of vessel branching in E14.5 cortex shown in **G**, n = 4. Scale bar: 200 μm.

The following figure supplement is available for figure 4:

**Figure supplement 1**. The *Nrp1^VEGF−^* mutant mice display normal vessel branching and coverage at postnatal stages.

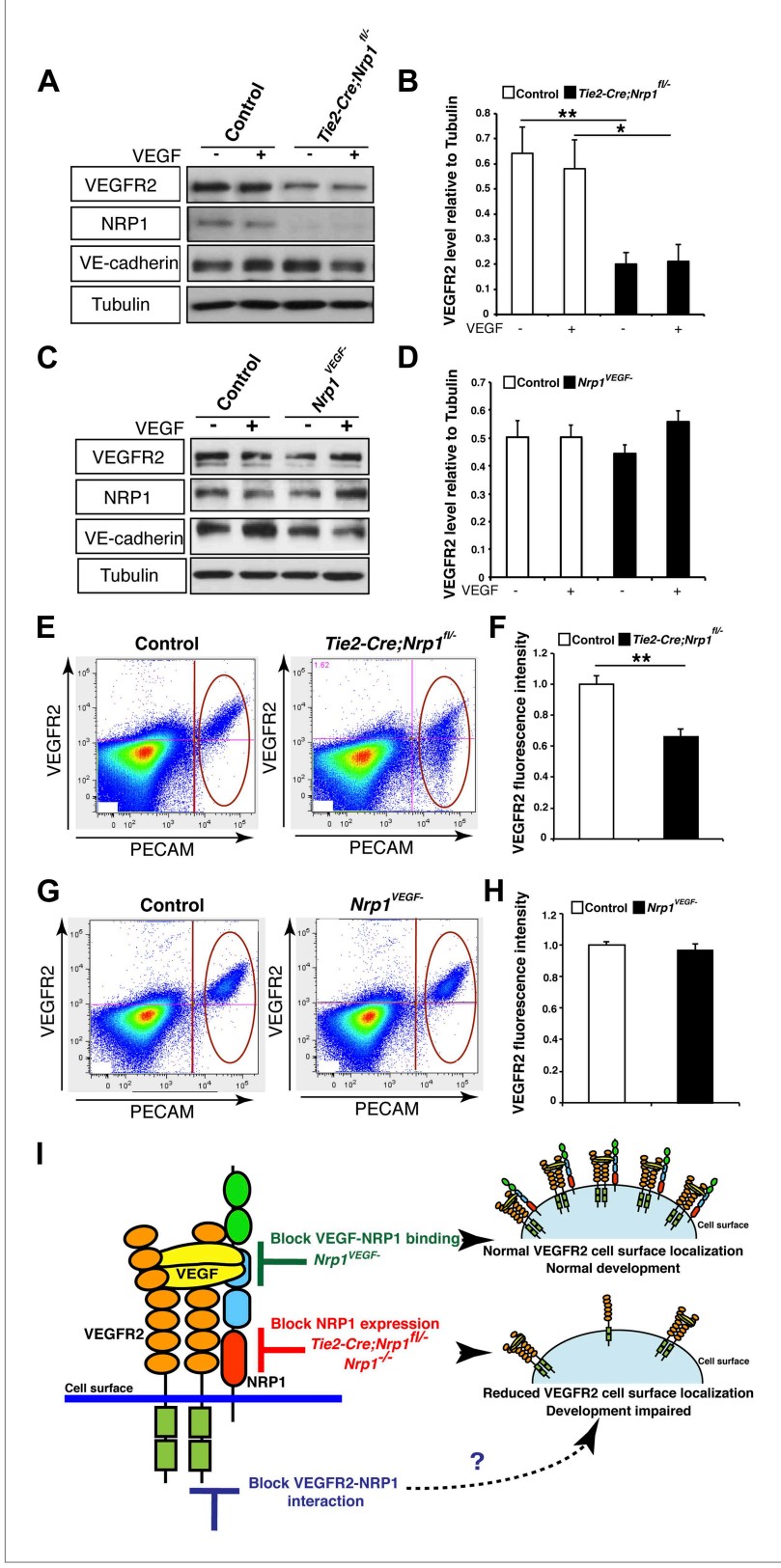

**Figure 5**. NRP1 regulates VEGFR2 expression at the cell surface independent of VEGF-NRP1 binding. (**A**) Western blot from E14.5 lung tissue treated with 50 ng/ml VEGF for 15 min revealed that VEGFR2 was reduced in *Tie2-CreNrp1*[fl/–] mutants while VE-cadherin expression remained at control levels. Western blot for NRP1 demonstrates that the
*Figure 5. Continued on next page*

*Figure 5. Continued*

*Tie2-Cre* allele successfully knocked down NRP1 expression. (**B**) Quantification of VEGFR2 expression shown in **A**, n = 4. (**C**) Western blot from E14.5 lung tissue treated with 50 ng/ml VEGF for 15 min demonstrates that VEGFR2, NRP1, and VE-cadherin expression were unperturbed in the *Nrp1^VEGF−^* mutants. (**D**) Quantification of VEGFR2 expression shown in **C**, n = 5. (**E**) FACS analysis plots illustrate a reduction in VEGFR2 surface expression in endothelial cells isolated from *Tie2-Cre;Nrp1^fl/−^* mice. (**F**) Quantification of the VEGFR2 fluorescence intensity from the FACS analysis shown in **E**, n = 5. (**G**) FACS analysis plots demonstrate that VEGFR2 surface expression in endothelial cells isolated from *Nrp1^VEGF−^* mice remained at control levels. (**H**) Quantification of the VEGFR2 fluorescence intensity from the FACS analysis shown in **G**, n ≥ 7. (**I**) Schematic of VEGFR2 and NRP1 at the cell surface illustrates VEGF ligand binding to both VEGFR2 and NRP1. In the *Nrp1^VEGF−^* mutants, VEGF-NRP1 binding is abolished, VEGFR2 has normal cell surface localization, and vascular development proceeds appropriately. However, in *Nrp1^−/−^* mutants, VEGFR2 cell surface localization is reduced and vascular development is impaired.

The following figure supplements are available for figure 5:

**Figure supplement 1**. VEGF-NRP1 binding is not required for NRP1-VEGFR2 complex formation in vitro and in vivo.

**Figure supplement 2**. VEGF-induced VEGFR2 phosphorylation is reduced in both the *Nrp1^VEGF−^* and *Tie2-Cre;Nrp1^fl/−^* mutants.

---

the vascular extension and artery number but did not have any abnormalities in vessel coverage as compared with control littermates (***Figure 6A***). In the adult, the vascular extension and vessel coverage in the retina were indistinguishable from controls (***Figure 6B***) indicating that the *Nrp1^VEGF−^* mutants experience a delay in the formation of the primary vascular plexus. However, the number of retinal arteries remained lower in *Nrp1^VEGF−^* adults. These results demonstrate that VEGF-NRP1 interactions are required to some degree for postnatal angiogenesis and artery differentiation in the retina.

In addition, *Nrp1^VEGF−^* animals were also assessed for injury-induced arteriogenesis following femoral artery ligation. In this assay, the femoral artery was surgically severed in both *Nrp1^VEGF−^* and control mice, and blood flow recovery was monitored via deep penetrating laser Doppler imaging. Femoral artery ligation produced a comparable level of hindlimb ischemia in the *Nrp1^VEGF−^* mutants and controls (***Figure 6—figure supplement 1***). However, the *Nrp1^VEGF−^* mutants exhibited a significant delay in hindlimb re-perfusion. Building upon these results, future work will utilize the *Nrp1^VEGF−^* knock-in line to determine if VEGF-NRP1 signaling functions in pathological or physiological angiogenesis in the adult.

## Discussion

In this study, we identified a single amino acid within the extracellular b1 domain of NRP1 that is required for VEGF-NRP1 binding, but non-essential for SEMA3-NRP1 interactions. A point mutation in this D320 residue was incorporated into the endogenous *Nrp1* locus to generate the *Nrp1^VEGF−^* mutant, a novel mouse line that selectively abolishes VEGF-NRP1 binding in vivo. Recently a cDNA knock-in NRP1 mutant, *Nrp1^Y297A/Y297A^*, was also developed to examine the role of VEGF-NRP1 binding (***Fantin et al., 2014***). However, mice generated with genetically modified cDNA notoriously lack the essential intronic regions that regulate the temporal and spatial expression of the gene. Consequently, the aberrant and severe down-regulation of NRP1 protein expression in the *Nrp1^Y297A/Y297A^* hypomorph prevents any definitive conclusions from being garnered about the biological cause of phenotypes present in this mouse. In this regard, abnormalities in the *Nrp1^Y297A/Y297A^* hypomorph could originate from two potential sources: the severe reduction in NRP1 levels or the abolishment of VEGF-NRP1 binding. Unlike the *Nrp1^Y297A/Y297A^* line, our *Nrp1^VEGF−^* mutant contains a two base pair replacement in the endogenous *Nrp1* locus and preserves the genetic structure of the *Nrp1* gene. Consequently, *Nrp1^VEGF−^* mice maintain appropriate levels of NRP1 protein expression and allow the first unobscured in vivo assessment of VEGF-NRP1 binding in developmental angiogenesis. Our *Nrp1^VEGF−^* line provides a powerful new genetic tool for selectively interrogating the function of VEGF-NRP1 binding in broad areas of basic research and translational study.

Remarkably, our *Nrp1^VEGF−^* mutant did not recapitulate the early embryonic lethality or developmental angiogenesis phenotypes of the *Nrp1^−/−^* and endothelial-specific NRP1 knock-out (***Figure 4***).

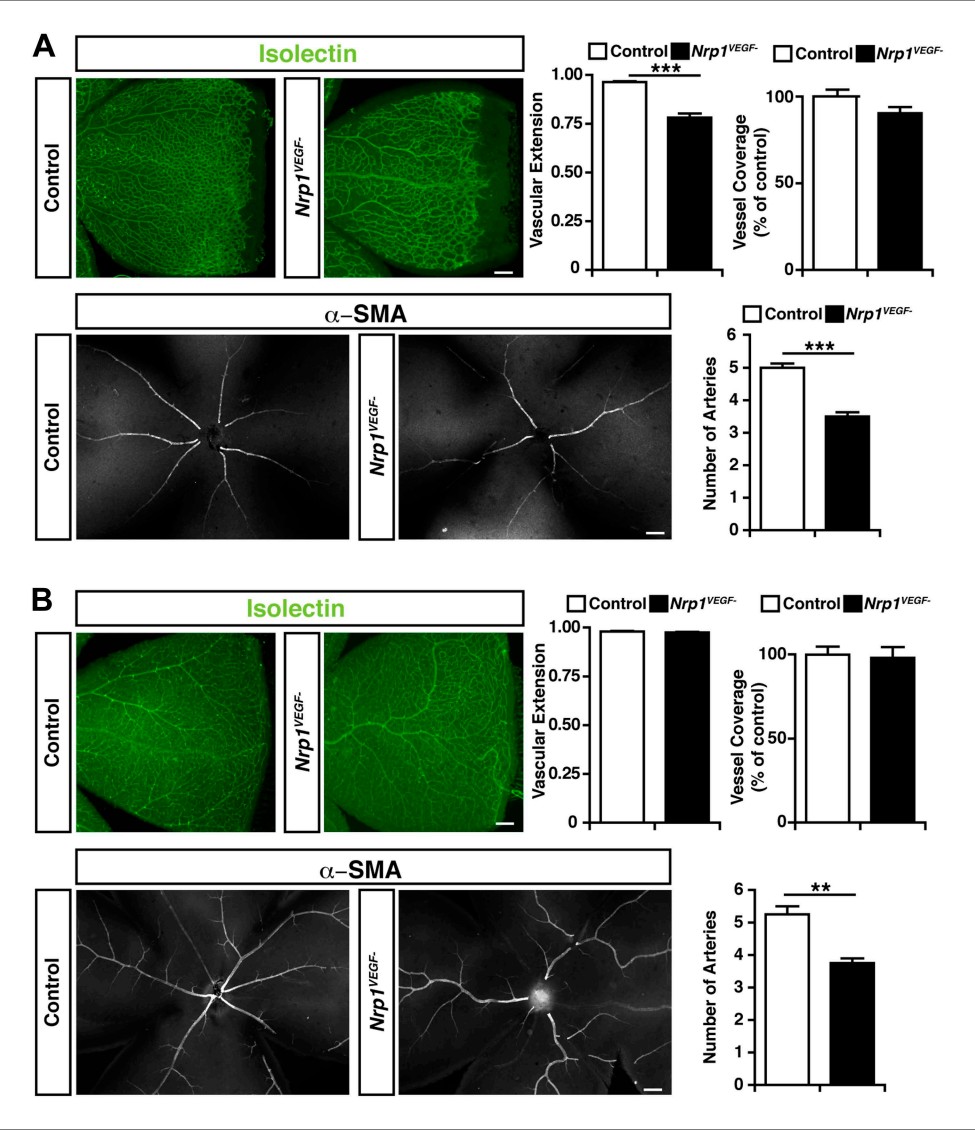

**Figure 6**. Retinal angiogenesis is perturbed in the *Nrp1^VEGF–* mutant. (**A**) Isolectin and α-SMA staining on P9 retinal flat-mounts revealed a significant reduction in vascular extension and artery number in *Nrp1^VEGF–* mutants. However, vessel coverage in the retina was unperturbed in the *Nrp1^VEGF–* mutants, n = 6. (**B**) In the adult, isolectin and α-SMA staining showed that the number of retinal arteries remained lower in the *Nrp1^VEGF–* mutants than littermate controls while vascular extension and vessel coverage in the retina were normal, n = 4. Scale bar: 200 μm.

The following figure supplement is available for figure 6:

**Figure supplement 1**. The *Nrp1^VEGF–* mutants have delayed blood flow recovery following femoral artery ligation.

Moreover, the *Nrp1^VEGF–* mutant did not exhibit any of the cardiac failure, perinatal lethality, or growth defects observed in the *Nrp1^Y297A/Y297A* hypomorph indicating that these phenotypes are attributed to the severe reduction in NRP1 protein in *Nrp1^Y297A/Y297A* mutants rather than the lack of VEGF-NRP1 binding. However, the *Nrp1^VEGF–* mutant did exhibit a delay in vascular extension and a reduction in the number of arteries in the postnatal retina. This retinal phenotype is significantly less severe than those observed in the *Nrp1^Y297A/Y297A* hypomorph (*Fantin et al., 2014*) or in animals treated with antibodies inhibiting VEGF-NRP1 binding (*Pan et al., 2007*). Together, these results reveal that the retina relies on both VEGF-NRP1 dependent and independent mechanisms to establish the retinal vasculature.

Our surprising results challenge the well-accepted view that NRP1 depends on VEGF-NRP1 binding to facilitate angiogenesis and points to a provocative new hypothesis that NRP1 functions independently of VEGF-NRP1 binding perhaps via its interaction with an unidentified ligand or in its capacity as a co-receptor for VEGFR2. Our study demonstrates that the NRP1-deficient endothelial cells have reduced VEGFR2 expression at the cell surface, a phenomenon that was not observed in the *Nrp1$^{VEGF-}$* mutants. This result provides the first in vivo evidence that NRP1 controls VEGFR2 levels at the cell membrane and offers the first in vivo phenotypic characterization linking NRP1 regulated VEGFR2 surface expression to vascular development.

Consistent with our in vivo observations, several lines of in vitro work using multiple cell culture systems demonstrate that NRP1 is essential for the proper presentation, recycling, and degradation of VEGFR2 (*Shintani et al., 2006*; *Holmes and Zachary, 2008*; *Ballmer-Hofer et al., 2011*; *Hamerlik et al., 2012*). The loss of function and gain of function studies in human umbilical vein endothelial cells (HUVECs) found that VEGFR2 protein levels were decreased in the absence of NRP1 while *Vegfr2* mRNA levels were unaffected by Nrp1 siRNA (*Shintani et al., 2006*; *Holmes and Zachary, 2008*). Similarly, *Hamerlik et al. (2012)* examined human glioblastoma multiforme cells and found that shRNA mediated knock-down of NRP1 resulted in dramatically decreased VEGFR2 protein levels accompanied by a lower surface presentation of VEGFR2 and a decrease in cell viability. Moreover, cell surface protein biotinylation and immunofluorescence staining with confocal microscopy confirmed the co-localization of VEGFR2-NRP1 with the early/recycling endosome. Finally, *Ballmer-Hofer et al., (2011)* used stably transfected porcine aortic endothelial cell (PAEC) lines in conjunction with immunostaining to visually follow VEGFR2 trafficking in the presence and absence of NRP1. Their experiments revealed that upon VEGF stimulation, VEGFR2 is internalized in Rab7 vesicles for degradation. However, in the presence of NRP1, VEGFR2 is stabilized in Rab11 vesicles and recycled back to the cell surface. In conjunction with our in vivo results, these data demonstrate that NRP1 guides vascular development through its capacity as a VEGFR2 co-receptor rather binding to VEGF. In this manner, NRP1 regulates angiogenesis by controlling the amount of VEGFR2 expression at the cell surface and consequently the level of VEGFR2-VEGF signaling.

The modulation of co-receptors may function as a general mechanism for regulating cell signaling and behavior. A prior in vitro study identified a similar relationship between the membrane protein, neural cell adhesion molecule (NCAM) and fibroblast growth factor receptor-1 (FGFR1) (*Francavilla et al., 2009*). This previous work discovered that NCAM induced sustained FGFR1 activation by controlling the intracellular trafficking of the FGFR1 receptor. Specifically, NCAM was capable of re-targeting internalized FGFR1 from the lysosomal degradation pathway to Rab11-postive recycling vesicles and increased FGFR1 expression at the cell surface. In this regard, the co-receptor interaction between NRP1 and VEGFR2 may be representative of a more universal phenomenon in which membrane proteins function to regulate the cell surface expression and subsequent downstream signaling of receptors.

Ultimately, our findings mark a pivotal step toward understanding the role of NRP1 in developmental angiogenesis and indicate that NRP1-VEGFR2 interactions rather than VEGF-NRP1 binding underlie NRP1's critical function in VEGF-mediated vascular development. Given the substantial resources invested in NRP1-targeted anti-angiogenesis therapies for vascular disease and cancer, the information gleaned from this study will be invaluable in identifying the cellular and molecular mechanisms underlying angiogenesis and ultimately using this information to instruct the development of new therapeutic approaches.

## Materials and methods

### Site-directed mutagenesis and targeting vector construction

Rat Neuropilin1 cDNA was re-cloned from pMT21 into pCS2+ using the original EcoRI and XhoI sites present in both vectors. Mutations were made using PCR, and the mutated fragment was subcloned back into pCS2-Nrp1 using endogenous restriction sites. The targeting vector (TV) was constructed using a combination of traditional cloning and recombineering along with point mutagenesis. Genomic DNA was obtained from the 129S7-AB2.2 BAC library, clone #bMQ-373E22. The short (3') arm (1.3 kb) was cloned into the HpaI and EcoRI sites of 4600C-loxP. Two short homology arms (900 bp, total) were created and cloned into the XhoI and NotI sites of 4600C-loxP, with the two arms joined by a SalI site. The homology arms were ligated in a triple ligation to 4600C-loxP as well as to each other. The vector

was then linearized with SalI and electroporated into modified electrocompetent DH10B cells containing the previously mentioned BAC in order to facilitate homologous recombination to insert the remainder of the long arm. Recombineering was performed as described by the NCI-Frederick. After a full-length TV was made, the D320K mutation was introduced. The final TV was linearized and electroporated into ES cells. All primer sequences used for the targeting vector construction are provided in *Supplementary file 1*.

## Alkaline-phosphatase-tagged ligand production

HEK293T cells were transfected with AP-SEMA3A, AP-VEGF A, AP-VEGF B, or AP-PlGF expression constructs using a calcium phosphate transfection method. Media was changed after 6 hr. Cells were cultured for an additional 48 hr in DMEM + 10% FBS. After 48 hr the media were collected, filtered to remove the cell debris, and AP activity was measured. The ligands were frozen at −80°C until use.

## Binding of AP-tagged protein to cells and unpermeabilized antibody staining

COS-1 cells were grown in DMEM + 10% fetal bovine serum (FBS) + 1% Penicillin-Streptomycin. Cells were transfected with the indicated expression vectors using Lipofectamine-2000 (Invitrogen, Carlsbad, CA) in 6-well plates. 24 hr later, transfected cells were split into 24-well plates for parallel AP-binding and antibody staining. 24 hr after splitting, binding was performed using AP-tagged ligands (AP-VEGF A, AP-SEMA3A, AP-VEGF B, AP-PlGF). The binding protocol was as follows: cells were washed 1× with HBHA (1× HBSS, 0.5 mg/ml BSA, 0.5% sodium azide, and 20 mM HEPES [pH 7]), then incubated for 75 min with 0.3 ml of 2 nM ligand. Cells were then washed 7× with HBHA on a rotating platform and 110 μl of cell lysis buffer (1% Triton X-100 and 10 mM Tris–HCl [pH 8]) was added to each well. Cells and buffer were scraped into Eppendorf tubes, then vortexed for 5 min to fully lyse them. The lysates were then spun down for 5 min, and the supernatant was heat inactivated at 65°C for 10 min to inactivate endogenous alkaline phosphatases. AP-activity was measured by adding 2× SEAP buffer (50 ml 2 M diethanolamine [pH 9.8], 50 μl 1 M $MgCl_2$, 224 mg L-homoarginine, 50 mg BSA, 445 mg p-nitrophenylphosphate) and measuring optical absorbance at 405 nm every 15 s for 1 min. Antibody staining of these cells was done as follows: non-specific binding was blocked with 5% Normal Goat Serum in DMEM for 30 min at 4°C. Cells were then incubated with primary antibody (Rabbit anti-NRP1, gift of Dr David Ginty) for 2 hr at 4°C. They were then washed 6× with cold HBHA, then incubated with a secondary antibody (AP-tagged anti-rabbit) for 1.5 hr at 4°C. Cells were then washed 3× in cold HBHA, then lysed as described above. AP-activity was measured from lysed extracts. Binding of AP-tagged ligands was normalized to protein content of each well and to antibody staining with an anti-NRP1 antibody. Each AP-binding assay was independently repeated three times.

## Animal care

$Nrp1^{VEGF-}$, Tie2-Cre, $Nrp1^{fl}$, and $Nrp1^-$ (*Gu et al., 2003*) mice were maintained on a C57Bl/6 background. $Nrp1^{VEGF-}$ mice were genotyped with traditional PCR techniques. The expected WT band is 305 bp, while the targeted allele is 350 bp due to the remaining presence of one FRT site. To sequence the mutation site, PCR was performed to generate a fragment around the mutation site. The primer sequences for genotyping and sequencing are included in *Supplementary file 1*. Tie2-Cre, $Nrp1^{fl}$, and $Nrp1^-$ genotyping was performed as previously published. All animals were treated according to institutional and NIH guidelines approved by IACUC at Harvard Medical School.

## AP-ligand binding to tissue sections

Embryos were dissected and frozen immediately in liquid nitrogen, then stored at −80°C until use. Sections were cut at 25 μm with a cryostat, then fixed for 8 min in ice-cold methanol. Sections were then washed 3× in PBS + 4 mM $MgCl_2$. Non-specific binding was reduced by blocking the sections with DMEM + 10% FBS for 45 min. After fixation, sections were incubated with 2 nM AP-tagged ligand, diluted with PBS + 4 mM $MgCl_2$, and buffered with HEPES, pH 7 for 1.5 hr at room temperature in a humidified chamber. The sections were washed 5× in PBS + 4 mM $MgCl_2$, then fixed with a fixative solution (60% acetone, 1% formaldehyde, 20 mM HEPES, pH 7). Sections were washed 3× in PBS and incubated in PBS at 65°C for 2 hr to heat inactive endogenous alkaline phosphatases and then incubated overnight in developing solution (100 mM Tris–HCl pH 9.5, 100 mM NaCl, 5 mM $MgCl_2$) with NBT (nitro-blue tetrazolium chloride) and BCIP (5-bromo-4-chloro-3'-indolyphosphate p-toluidine). AP-ligand binding was analyzed in sections from at least three animals across two different litters per genotype.

## Western blotting

For immunoblotting, E14.5 lung samples were loaded on 8% polyacrylamide gels and run until the appropriate protein separation was achieved. Samples were electrophoretically transferred onto the PVDF membrane. Non-specific binding was blocked by a 1 hr incubation in 5% non-fat milk in TBST (Tris-buffered saline + 0.1% Tween-20). The membranes were then incubated overnight with the following primary antibodies, as indicated below, at 4°C: anti-NRP1 (#ab81321 Abcam, Cambridge, MA or gift of Dr David Ginty, see *Ginty et al., 1993* for details), anti-VEGFR2 (gift of Procter and Gamble, see *Gu et al., 2003* for details), anti-VE-cadherin (#ab33168 Abcam, Cambridge, MA), anti-p-VEGFR2 (p1175) (#2478 Cell Signaling Technology, Danvers, MA), and anti-α-Tubulin (#T5168 Sigma-Aldrich, Natick, MA). After incubation with primary antibodies, the membranes were washed 3× in TBST then incubated with the appropriate HRP-labeled secondary antibody in TBST or 5% milk in TBST for 1 hr at room temperature. Membranes were then washed 3× with TBST then developed with regular or super ECL (GE Amersham, United Kingdom or Thermo Scientific, Waltham, MA). The intensity of individual bands was quantified using ImageJ.

## Phenotypic analysis of the *Nrp1^VEGF−* mutant

At the indicated stages, embryos were dissected, fixed with 4% paraformaldehyde, equilibrated in a sucrose gradient, embedded in OCT, and sectioned in the coronal plan at 12 μm with a Leica CM3050S cryostat. Likewise, the brains of postnatal pups (P7) were dissected, fixed, cryo-protected, and sectioned at 20 μm. Tissue sections were washed 3× for 5 min in 0.2% PBT (0.2% Triton X-100 in PBS), incubated with Isolectin GS-IB4 (#I21411 Life Technologies, Grand Island, NY) overnight at 4°C, washed 3× for 5 min in PBS, and coverslipped with using ProLong Gold/DAPI antifade reagent (#P36935 Molecular Probes, Eugene, OR). Sections were imaged by fluorescence microscopy using a Nikon Eclipe 80i microscope equipped with a Nikon DS-2 digital camera. Quantification was performed using ImageJ. Vessel coverage delineates the percent of cortical pixel area covered by isolectin-positive pixels while vessel size quantifies the pixel area of each discrete vascular aggregate identified by isolectin staining.

## VEGF lung treatment

E14.5 mouse lungs were dissected in cold PBS and minced finely using a razor blade. The tissue was then incubated with plain EBM (Lonza, Switzerland) or EBM containing 50 ng/ml VEGF for 15 min at 37°C. Lysis buffer (50 mM Tris/HCl [pH 7.5], 150 mM NaCl, 1% Triton X-100, 2 mM EDTA, and 2 mM DTT) containing complete proteinase inhibitors (Roche, Switzerland), PhosSTOP (Roche, Switzerland), and sodium orthovanadate was added to the tissue, which was then pulverized with a pestle and incubated for 30 min while rotating at 4°C. Tissue was spun down and protein quantification was performed. The tissue was treated as described in the Western blotting section.

## Co-immunoprecipitation

HEK293T cells were transfected with the indicated constructs using Lipofectamine-2000 (Invitrogen, Carlsbad, CA). They were then grown in DMEM + 10% fetal bovine serum + 1% Penicillin-Streptomycin and 48 hr after transfection cells were washed and harvested in ice-cold PBS. Cells were lysed using lysis buffer (50 mM Tris/HCl [pH 7.5], 150 mM NaCl, 1% Triton X-100, 2 mM EDTA, and 2 mM DTT) containing complete proteinase inhibitors (Roche, Switzerland). After 30 min of rotation in the cold room and subsequent centrifugation, protein was quantified and 20 μg of protein was frozen down as input controls. 0.5 μg of anti-VEGFR2 antibody (gift of Procter and Gamble, see *Gu et al., 2003* for details) was added to 500 μg of protein and rotated in the cold room for 1 hr. Then, 20 μl of protein A/G beads (Thermo Scientific, Waltham, MA) were added to the protein and rotated overnight in the cold room. Beads were washed 3× with lysis buffer and two times with wash buffer (lysis buffer with 300 mM NaCl). Protein was eluted by the addition of 2× SDS-PAGE sample buffer and boiling for 10 min. Co-immunoprecipitation was also performed on P7 lung lysates isolated from control and *Nrp1^VEGF−* animals treated with VEGF as described above.

## FACS

Analysis of E14.5 mouse embryos were performed on single cells from dissociated lungs. In brief, microdissection techniques were used to isolate the lung. Lungs were then rinsed in PBS and incubated in 2 mg/ml collagenase and 20 μg/ml DNase I 3× for 15 min at 37°C and gently pipetted. The collagenase was inactivated using 5 ml of ice-cold 10% FBS/PBS, centrifuged at 1000×*g* for 5 min, and suspended in 400 μl of red blood cell (RBC) lysis buffer (Sigma-Aldrich, Natick, MA). Following a 5 min incubation

at room temperature, 2 ml of ice-cold 5% FBS/PBS was added and cells were centrifuged at 1000×$g$ for 5 min at 4°C. Cells were then blocked in Fc-blocking solution (#553142; BD) for 20 min on ice, centrifuged, incubated with the labeled conjugated primary antibodies–PE-anti-CD31 (PECAM) (#553373 BD Pharmingen, Franklin Lakes, NJ) and APC-anti-Flk1-1 (VEGFR2) (#560070 BD Pharmingen, Franklin Lakes, NJ), for 30 min on ice with agitation every 10 min. After incubation, the cells were spun down, the supernatant was removed, and the cell pellet was resuspend in 1:10K Sytox in PBS/5%FBS. Cells were analyzed on a LSR II Flow Cytometer. Cells incubated with no antibody, APC-anti-Flk1, or PE-anti-CD31 only served as the control population.

## Phenotypic analysis of the developing retina

Whole-mount retina immunohistochemistry was performed as previously described in *Kim et al., (2011)*. Briefly, eyes were extracted and fixed in 4% paraformaldehyde for 10 min at room temperature. Retinas were dissected in PBS and post-fixed in 4% paraformaldehyde overnight at 4°C. Retinas were then permeabilized in PBS, 1% BSA, and 0.5% Triton X-100 at 4°C overnight, washed 2× for 5 min in 1% PBT (1% Triton X-100 in PBS), and incubated in Isolectin GS-IB4 (1:200, #I21411 Life Technologies, Grand Island, NY) and anti-αSMA Cy3 (1:100, #C6198 Sigma-Aldrich, Natick, MA) in 1% PBT overnight at 4°C. Retinas were washed 3× for 5 min and flat-mounted using ProLong Gold antifade reagent (#P36934 Molecular Probes, Eugene, OR). Flat-mounted retinas were analyzed by fluorescence microscopy using a Nikon Eclipe 80i microscope equipped with a Nikon DS-2 digital camera and by confocal laser scanning microscopy using an Olympus FV1000 confocal microscope. Quantification was performed using MetaMorph Image Analysis Software and ImageJ. At least four retinal leaves were quantified per animal to determine the vascular extension ratio, both eyes were examined in each animal for artery number, and three representative images were quantified from each animal for vascular coverage (representing the total isolectin-positive pixel area per image).

## Femoral artery ligation

Ketamine (80–100 mg/kg) and xylazine (5–10 mg/kg) delivered by IP injection were used to anesthetize 12-week old male *Nrp1*$^{VEGF−}$ and control littermates. After anesthesia was achieved, the bilateral hindlimbs and lower abdomen were cleared of hair and cleaned with 10% betadine and 70% alcohol. An incision of 3–4 mm was made in the right inguinal area to visualize the femoral artery. Two 6–0 silk sutures were tied in the proximal femoral artery and the deep femoral and epigastric artery branches were cauterized. The femoral artery was then ligated between the two sutures. The skin was sutured with one 4–0 prolene sutures. Immediately before and after surgery, each animal was scanned with a non-invasive laser doppler imaging system (moorLD12-HR Moor Instruments, Wilmington, DE) under 1–3% isofluorane anesthesia. Blood flow recovery in the hindlimbs was further assessed on 3, 5, and 7 days post-surgery and quantified via Moor LDI Software.

## Statistical analysis

The standard error of the mean was calculated for each experiment and error bars in the graphs represent the standard error. A paired Student's *t*-test was used to determine the statistical significance of differences between samples, and the genotype distribution was analyzed using a Chi-square test. Statistical analyses were performed with Prism 4 (GraphPad Software) and p values are indicated by * $\leq$ 0.05, ** $\leq$ 0.01, and *** $\leq$ 0.001.

## Acknowledgements

We thank the members of the Gu laboratory for helpful comments on the manuscript, Lauren Byrnes for technical support, and both the Flow Cytometry Facility in the Systems Biology Department and the Neurobiology Imaging Facility in the Neurobiology Department of Harvard Medical School for consultation and instrument availability that facilitated this work. The Neurobiology Imaging Facility is supported in part by the Neural Imaging Center as part of an NINDS P30 Core Center grant #NS072030. This study was supported by the National Institutes of Health Fundamental Neurobiology Training grant T32 NS007484-12 (N Hagan), the Alice and Joseph Brooks Fund Postdoctoral Fellowship (A Tata), Harvard Mahoney Neuroscience Institute Fund Postdoctoral Fellowship (B Lacoste), National Institutes of Health grant R01 HL096384 (K Kang and J Bischoff), and the following grants to C Gu: Sloan research fellowship, Armenise junior faculty award, the Genise Goldenson fund, and National Institutes of Health grant R01 NS064583.

## Additional information

### Funding

| Funder | Grant reference number | Author |
|---|---|---|
| National Institutes of Health | R01 NS064583 | Chenghua Gu |
| Harvard Medical School | Alice and Joseph Brooks Fund Postdoctoral Fellowship | Aleksandra Tata |
| Harvard Medical School | Harvard Mahoney Neuroscience Institute Fund Postdoctoral Fellowship | Baptiste Lacoste |
| Alfred P. Sloan Foundation | Sloan Research Fellowships | Chenghua Gu |
| Giovanni Armenise-Harvard Foundation | Armenise Junior Faculty Award | Chenghua Gu |
| Harvard Medical School | Genise Goldenson Research Fund | Chenghua Gu |
| National Institutes of Health | R01 HL096384 | Kyu-Tae Kang, Joyce Bischoff |
| National Institutes of Health | T32 NS007484-12 | Nellwyn Hagan |

The funders had no role in study design, data collection and interpretation, or the decision to submit the work for publication.

### Author contributions

MVG, NH, AT, W-JO, Conception and design, Acquisition of data, Analysis and interpretation of data, Drafting or revising the article; BL, K-TK, Acquisition of data, Analysis and interpretation of data, Drafting or revising the article; JK, Acquisition of data, Drafting or revising the article, Contributed unpublished essential data or reagents; JB, CG, Conception and design, Analysis and interpretation of data, Drafting or revising the article; J-HW, Analysis and interpretation of data, Drafting or revising the article, Contributed unpublished essential data or reagents

### Ethics

Animal experimentation: This study was performed in strict accordance with the recommendations in the Guide for the Care and Use of Laboratory Animals of the National Institutes of Health. All animals were handled according to approved institutional animal care and use committee (IACUC) protocols at Harvard Medical School (IACUC Study ID: IS00000045).

## Additional files

### Supplementary file

• Supplementary file 1. Primers used for generating, genotyping, and sequencing the $Nrp1^{VEGF-}$ knock-in mouse line.

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
