## [Decision Letter]

Thank you for sending your work entitled “Neuropilin-1 controls developmental angiogenesis by regulating VEGFR2 independent of VEGF-Neuropilin-1 binding” for consideration at *eLife.* Your article has been favorably evaluated by Janet Rossant (Senior editor), Jeremy Nathans (Reviewing editor), and 3 reviewers.

The manuscript has been reviewed by three expert reviewers, and their assessments together with the Reviewing editor form the basis of this letter. I am also including the three reviews in their original form at the end of this letter, as there are many specific and useful suggestions in them that will not be repeated in the summary here.

All of the reviewers were impressed with the incisive design and execution of your experiments. The mutant mouse that you constructed is the cleanest experimental approach to date to address the question of whether direct VEGF binding to Neuropilin1 (Nrp1) plays a role in vivo. Our consensus is that this is an important body of work but that a few key questions have been left open and that addressing them would greatly enhance the impact of the study.

These are:

Is the Nrp1 VEGF binding mutant mouse really normal? A close look at the retinal vasculature (e.g. at P7, P9, and adulthood) and at health/fertility at later ages would be useful. You might consider stressing the system, for example with the oxygen induced retinopathy model (your colleague Lois Smith at Children's Hospital is the expert) or with a test of tumor angiogenesis, both of which tie into the applied aspects of this work as related to the efficacy of anti-Nrp1 antibodies. Comparing the phenotypes to those reported by [6] would be useful.

The decrease in VEGFR2 levels (and surface VEGFR2), which is presented as the mechanism by which the Nrp1 KO exerts its effects, could use more experimental backing. Checking the level of VEGFR2 phosphorylation in response to VEGF and an assessment of VEGFR2 trafficking with vs. without Nrp1 vs. with the Nrp1 VEGF binding mutant (in cell culture) would be very helpful.

Is the VE cadherin level in Figure 5 lower in the absence of Nrp1? If so, does this suggest other roles for Nrp1? Also, the decrease in VEGFR2 by Western blot seems more dramatic than the decrease shown by FACS. Is the lower level of VEGFR2 shown by FACS in the absence of Nrp1 really insufficient for VEGF signaling? Or is something more subtle at play (defective trafficking?)?

Reviewer #1:

Clear mechanistic understanding of the VEGF signaling pathway is of paramount importance since this pathway lies in the heart of vascular biology and is being targeted to treat a number of human diseases. The role of Neuropilin-1 is an important piece of the puzzle in VEGF signaling, and the central dogma about Neuropilin-1 is that binding to VEGFA is essential for its function despite lacking direct evidence to support it. Recent attempt to provide direct evidence by assessing a Neuropilin-1 allele that lacks VEGF-A binding capacity fell short to answer this question due to the severe reduction of expression (6). In this report, Gelfand et al carried out rigorous analyses of a knockin Neuropilin-1 allele that abolished VEGFA binding yet preserved expression levels, and provided convincing evidence to demonstrate that the VEGFA binding capacity is dispensable for embryonic vascular development. They also provided biochemical evidence implicating Neuropilin-1 in the regulation of VEGFR2 levels. This study unequivocally refutes a dogma in the VEGF signaling field and reveals a novel Neuropilin-1 function. It is suitable for publication in *eLife*. The overall quality of the paper can be improved by addressing the following comments.

1) The degree of surface VEGFR2 reduction is much less than that of total VEGFR2 when Npn1 was deleted in ECs (compare Figure 5 with 5A), suggesting that the loss of Npn1 effect on surface VEGFR2 might be secondary to a reduction of total VEGFR2. Since the exiting data cannot distinguish between maintaining surface VEGFR2 levels versus regulating total VEGFR2 levels, the mechanism depicted in Figure 5 is not supported by actual data. The diagram should be modified to reflect a reduction in total VEGFR2, and the related text descriptions throughout the manuscript should be revised accordingly.

2) Since the reduction of surface VEGFR2 is modest (∼ 30%, Figure 5) in the Npn1 null ECs, the functional significance of reducing VEGFR2 levels has not been clearly established. This question can be addressed by including a phospho-VEGFR2 blot or ELISA in Figure 5.

3) Data from [6] indicated that either severely reduced NRP1 levels and/or abolishment of NRP1-VEGFA binding plays an important role in postnatal angiogenesis. This is an important issue in the angiogenesis field. The authors have the proper tool and data to address this question. They can contribute to the angiogenesis community by mentioning some aspects of postnatal development in their Npn1VEGFR- mice. For example, they can include a table that documents mouse numbers at different age after birth to demonstrate if there is reduced viability overtime, instead of only giving a statement about a single time point. They can also point out if they have seen overt phenotypes reported by [6], for example, have they ever seen evidence of cardiac failure in some of their knockin mice?

4) Some mechanistic data related to how NPN1 regulates VEGFR2 level will significantly elevate this manuscript to a different level. The authors should consider including one or two in vitro analysis similar to the NCAM / FGFR1 study (10).

Reviewer #2:

Gelfand et al present evidence that a point mutation (D320K) in the extracellular domain of neuropilin 1 (Npn1) prevents its binding to VEGF while leaving relatively intact Nrp1 association with Sema 3A. By gene replacement strategy they also generated a mouse line harboring the D320K mutation in the Npn1 locus.

The major finding of this paper is that, in absence of VEGF binding to Npn1, embryo vascular development is unaffected. Furthermore, it is reported that Npn1 co-immunoprecipitates with VEGFR2 and stabilizes VEGFR2 levels. The authors therefore hypothesize that Npn1 acts by controlling VEGFR2 levels through a still undefined mechanism.

1) The authors are not giving a fair representation of the literature. In the Abstract they report: “Npn1 is essential for vascular morphogenesis, but how Npn1 functions to guide vascular development remains elusive”. This statement is unfair considering several previous papers: Lanahan et al Dev Cell 2013; Fantin et al Development 2014 or Raimondi et al J Exp Med 2014. Furthermore, similarly to what is reported here, [6] already generated mice expressing Npn1 with a mutation that prevents VEGF binding. Homozygous mutated mice were viable and no severe and lethal cardiovascular phenotypes were observed implying that Npn1 could have VEGF-independent role during vascular development. Although the experimental approaches used in the two papers are different, the final conclusion remains the same.

2) The reduction in VEGFR2 levels in Npn1 mutants is indeed striking but does not correspond to the findings of many other authors where the uncoupling or strong reduction on Npn1 did not modify VEGFR2 or VEGFR1 expression (see, as an example, Fantin et al. Development 2014). In addition, while in Figure 5 the drop in receptor expression is almost complete; in Figure 5 it is around 35% only.

Since Title, Abstract and Discussion are focused on this finding the authors should present more solid data to support their conclusions. Last but not least, if the decrease in VEGFR2 is in the range presented in Figure 5 the phenotype of Npn1 KO should be comparable to that of VEGFR2 KO and this is not the case.

3) The authors do not investigate the possible effects of their Npn1 mutant on neurons. Taking into account that some types of neurons rely on VEGF165 and Npn1 signaling, these observations are needed for a more complete interpretation of the mutant phenotype.

4) Introduction. Most of previously published data converge in saying that the domain of Npn1 responsible for the binding of Sema 3a is the b1 region and not the a1 (see for instance Figure 2 Nat. Rev. Immunol. 2103 by Kikutani). This should be introduced and discussed in a better way in the text.

In conclusion, the model presented is of particular value and the data shown demonstrate unequivocally that Npn1 acts independently from binding to VEGF 165. However, more solid data on the mechanism of action of Npn1 are needed to make the paper novel enough for publication in the Journal.

Reviewer #3:

In this study the VEGF binding site of Neuropilin has been mutated in mice. This has been executed very well, with careful in vitro validation of the mutation to ensure that the mutation (D320K) only affects VEGF but not Sema3 binding. This approach is more sophisticated and better controlled than a previous study from another group who created a Nrp1Y297A mutation (PMID: 24401374), also aimed at ablating VEGF binding in Nrp1. The NrpY297A study suffered from a poor genetic targeting strategy that resulted in reduced Nrp1 expression (in vivo). Surprisingly, both studies found that VEGF binding to Nrp1 is not required for normal embryonic development. The current study shows this more convincingly, because the Nrp1Y297A study was more difficult to interpret because of the reduced Nrp1 levels caused by the Nrp1Y297A mutation. The novel Nrp1D320K strain described here will therefore be a very useful, well validated research tool.

Furthermore, the authors found that the Nrp1D320K mutation caused a dramatic reduction of VEGFR expression. This is a very intriguing observation and it would be interesting to know the reason for this and what the functional consequences are. The study does not explore this but the authors conclude (in the Discussion) that “Nrp1 regulates angiogenesis by controlling the amount of VEGFR2 expression at the cell surface and consequently the level of VEGFR2-VEGF signaling.”

There are several problems with this statement:

1) It is logically wrong. Since the Npn1VEGF mice have reduced VEGFR2 levels but develop normally, Nrp1 obviously is not regulating angiogenesis via VEGFR2 levels, otherwise the mice would have a Nrp1 KO phenotype.

2) The authors have not actually shown that VEGFR2 signalling is reduced. This would have to be shown more directly (e.g. VEGFR2 phosphorylation in Western blots).

3) Although the flow cytometry data in Figure 5 indicates reduced surface expression of VEGFR2 this seems only to be the case for around 50% of the cells. On the other hand, the Western blot data (Figure 5) indicates a much stronger reduction of around 90% (contradicting the flow cytometry data somewhat). This demonstrates that also intracellular VEGFR2 must be dramatically reduced and not only surface VEGFR2

4) VEGFR2 levels were only studied in lung endothelial cells. Yet the main vascular phenotype in Nrp1 mutants is found in the brain and retinal vasculature.

5) The Nrp1Y297A mice also developed normally but that study found abnormalities in postnatal animal, such as abnormal retinal vasculature development, reduced neovascularisation in the oxygen induced retinopathy model and reduced tumour growth. These postnatal phenotypes should be also checked in the Nrp1D320K mice.

In summary, this study has been executed very nicely but the main drawback is that it ends prematurely. When I got to Figure 5 I was expecting several more figures looking into receptor distribution/signalling in cultured cells etc. and in vivo phenotypes under stress conditions.

---

## [Author Response]

Is the Nrp1 VEGF binding mutant mouse really normal? A close look at the retinal vasculature (e.g. at P7, P9, and adulthood) and at health/fertility at later ages would be useful. You might consider stressing the system, for example with the oxygen induced retinopathy model (your colleague Lois Smith at Children's Hospital is the expert) or with a test of tumor angiogenesis, both of which tie into the applied aspects of this work as related to the efficacy of anti-Nrp1 antibodies. Comparing the phenotypes to those reported by [6] would be useful.

We thank the reviewers for this valuable suggestion. We have performed new experiments and now provide additional data describing the phenotype of the *Nrp1*^*VEGF-*^ mutant.

1) We generated a viability table detailing the survival of *Nrp1*^*VEGF-*^ mutants in comparison to their littermates across several different developmental stages (E14.5, P7 and adult, Figure 3—figure supplement 2). In contrast to the *Nrp1*^*Y297A/Y297A*^ hypomorph, *Nrp1*^*VEGF-*^ mutants are present at appropriate Mendelian ratios and exhibit a normal survival rate.

2) We grossly examined several organ systems (brain, heart, lung, and kidney) at P9 and adulthood and now include organ weights in Figure 3—figure supplement 2. Together with the whole mount images and body weight graph presented in Figure 3, our results clearly demonstrate that *Nrp1*^*VEGF-*^ mutants experience normal body growth.

3) We also generated whole mount images to illustrate the normal morphology of the *Nrp1*^*VEGF-*^ heart at P9 (Figure 3—figure supplement 2) and noted in the text that *Nrp1*^*VEGF-*^ mutants do not have any of the cardiac abnormalities observed in the *Nrp1*^*Y297A/Y297A*^ hypomorph.

4) We have also noted in the text that *Nrp1*^*VEGF-*^ mutants are fertile (pg 8) and in contrast to the *Nrp1*^*Y297A/Y297A*^ hypomorph, *Nrp1*^*VEGF-*^ mutants maintained appropriate NRP1 protein levels in several adult organ systems (Figure 3—figure supplement 2).

5) We examined the retinal vasculature in the *Nrp1*^*VEGF-*^ mutant at P9 and adulthood (Figure 6). Through this analysis, we uncovered that *Nrp1*^*VEGF-*^ mutants have normal vessel density and do not develop the endothelial tufts observed in the *Nrp1*^*Y297A/Y297A*^ mutant. *Nrp1*^*VEGF-*^ mutants do exhibit some of the angiogenesis phenotypes previously reported in [28] and [6] such as reduced vascular extension and reduced retinal arteries at P9. This mild retinal phenotype suggests that the retinal vasculature is uniquely sensitive to the loss of VEGF-NRP1 binding. Interestingly, the vascular extension phenotype is resolved in adulthood and the retinal abnormalities in the *Nrp1*^*VEGF-*^ mutant are substantially less severe than those described in [6] suggesting that some of the vascular defects reported in the *Nrp1*^*Y297A/Y297A*^ mutant are secondary to aberrant NRP1 protein levels rather than a lack of VEGF-NRP1 binding.

6) Finally, we performed femoral artery ligation surgeries to induce hind limb ischemia in *Nrp1*^*VEGF-*^ mutants and control littermates. In this challenge situation, the *Nrp1*^*VEGF-*^ mutant initially exhibited reduced blood flow recovery to the injured hind limb post, but eventually caught up to control levels by seven days post-surgery (Figure 6—figure supplement 1).

*The decrease in VEGFR2 levels (and surface VEGFR2), which is presented as the mechanism by which the Nrp1 KO exerts its effects, could use more experimental backing. Checking the level of VEGFR2 phosphorylation in response to VEGF and an assessment of VEGFR2 trafficking with vs. without Nrp1 vs. with the Nrp1 VEGF binding mutant (in cell culture) would be very helpful*.

We agree with these sentiments and have examined the level of VEGFR2 phosphorylation in lung lysates following VEGF treatment in the *Tie2-Cre,Nrp1*^*fl/-*^ and *Nrp1*^*VEGF-*^ mutants. We discovered that *Nrp1*^*VEGF-*^ mutants have a modest reduction in VEGFR2 phosphorylation at the tyrosine residue 1175 (Y1175) upon VEGF treatment without a change in total VEGFR2 protein (Figure 5; Figure 5—figure supplement 2). In contrast, the *Tie2-Cre;Nrp1*^*fl/-*^ mutant exhibits a significant reduction in VEGFR2 phosphorylation and VEGFR2 protein levels (Figure 5; Figure 5—figure supplement 2).

In regards to the suggestion for cell culture experiments, we have found several *in vitro* studies using multiple cell culture systems already examining the impact of NRP1 on VEGFR2 trafficking that demonstrate how NRP1 is essential for the proper presentation, recycling, and degradation of VEGFR2.

1) [31] used siRNA knock-down and adenovirus infection of *Nrp1* in human umbilical vein endothelial cells (HUVEC) to show that VEGFR2 protein level is decreased in the absence of NRP1 while *Vegfr2* mRNA levels are unaffected by NRP1 siRNA. Moreover, the overexpression of *Nrp1* led to an increase in VEGFR2 protein level as well as VEGFR2 cell surface expression.

2) Similarly, Holmes et al., 2008 found that NRP1 siRNA consistently and significantly reduced VEGFR2 expression upon VEGF induced receptor-mediated endocytosis, but found no significant changes in *Vegfr2* mRNA levels.

3) [1] used porcine aortic endothelial cell (PAEC) lines stably expressing VEGFR2, NRP1, or both in conjunction with immunostaining to visually follow VEGFR2 trafficking in the presence and absence of NRP1. In PAECs expressing only VEGFR2, VEGF stimulation triggered VEGFR2 to be internalized and accumulate in Rab7 vesicles for degradation. However, in the presence of NRP1, VEGFR2 is stabilized in Rab11 vesicles to be recycled back to the cell surface.

4) [14] examined human glioblastoma multiforme cells isolated from patients and found that shRNA mediated knock-down of NRP1 resulted in dramatically decreased VEGFR2 protein levels accompanied by a lower surface presentation of VEGFR2 and a decrease in cell viability. Furthermore, cell surface protein biotinylation and immunofluorescence staining with confocal microscopy confirmed the co-localization of VEGFR2-NRP1 with the early/recycling endosome

These *in vitro* studies are consistent with our *in vivo* results and provide clear mechanistic data related to how NRP1 regulates VEGFR2. Our current study provides the first *in vivo* evidence that NRP1 controls VEGFR2 levels at the cell membrane and offers the first *in vivo* phenotypic characterization linking NRP1 regulated VEGFR2 surface expression to vascular development.

*Is the VE cadherin level in*
Figure 5
*lower in the absence of Nrp1? If so, does this suggest other roles for Nrp1? Also*, *the decrease in VEGFR2 by Western blot seems more dramatic than the decrease shown by FACS. Is the lower level of VEGFR2 shown by FACS in the absence of Nrp1 really insufficient for VEGF signaling? Or is something more subtle at play (defective trafficking?)?*

To further clarify the western blot data shown in Figure 5, we have quantified the intensity of the bands from all of our blots and determined that VE-cadherin levels are not significantly reduced relative to Tubulin (see quantification below). In addition, we have quantified the total VEGFR2 protein levels in our Western blots (Figure 5) and selected a more representative blot to reflect this quantification. We also examined VEGFR2 phosphorylation in the *Tie2-Cre,Nrp1*^*fl/-*^ mutants to determine if the reduction in VEGFR2 levels in the absence of NRP1 is sufficient to alter VEGF signaling. Indeed, we found a significant reduction in VEGFR2 phosphorylation (almost absent) and included them in Figure 5—figure supplement 2.

Reviewer #1:

*1) The degree of surface VEGFR2 reduction is much less than that of total VEGFR2 when Npn1 was deleted in ECs (compare*
Figure 5
*with 5A), suggesting that the loss of Npn1 effect on surface VEGFR2 might be secondary to a reduction of total VEGFR2. Since the exiting data cannot distinguish between maintaining surface VEGFR2 levels versus regulating total VEGFR2 levels, the mechanism depicted in*
Figure 5
*is not supported by actual data. The diagram should be modified to reflect a reduction in total VEGFR2, and the related text descriptions throughout the manuscript should be revised accordingly*.

To reconcile the level of total versus surface VEGFR2 protein, we have gone back and quantified the VEGFR2 levels by Western blot and found that the reduction in VEGFR2 is more in line with the results observed in the FACS analysis. We have updated Figure 5, to include a more representative Western blot and a graph quantifying these results. Moreover, since the decrease in total VEGFR2 is comparable to the reduction in surface VEGFR2 seen via FACS analysis, we believe our model accurately reflects the data presented in this manuscript now.

*2) Since the reduction of surface VEGFR2 is modest (∼ 30%,*
Figure 5*) in the Npn1 null ECs, the functional significance of reducing VEGFR2 levels has not been clearly established. This question can be addressed by including a phospho-VEGFR2 blot or ELISA in*
Figure 5.

This point is a great suggestion, and we have performed experiments to quantify phospho-VEGFR2 upon VEGF treatment. In particular, we have examined the level of VEGFR2 phosphorylation in lung lysates following VEGF treatment in the *Tie2-Cre,Nrp1*^*fl/-*^ and *Nrp1*^*VEGF-*^ mutants.

*3) Data from*
[6]
*indicated that either severely reduced NRP1 levels and/or abolishment of NRP1-VEGFA binding plays an important role in postnatal angiogenesis. This is an important issue in the angiogenesis field. The authors have the proper tool and data to address this question. They can contribute to the angiogenesis community by mentioning some aspects of postnatal development in their Npn1VEGFR- mice. For example, they can include a table that documents mouse numbers at different age after birth to demonstrate if there is reduced viability overtime, instead of only giving a statement about a single time point*. *They can also point out if they have seen overt phenotypes reported by*
[6]*, for example, have they ever seen evidence of cardiac failure in some of their knockin mice?*

We thank the reviewer for this suggestion. Indeed, we have the proper *in vivo* tools to delineate whether the phenotypes described in [6] originate from the severe reduction in NRP1 levels or the abolishment of VEGF-NRP1 binding. We have expanded the phenotypic analysis of our *Nrp1*^*VEGF-*^ mutant by including a table that documents survival at different developmental stages to demonstrate the sustained viability of the *Nrp1*^*VEGF-*^ mutant. We have also examined organ weight, heart morphology, and retinal angiogenesis. As described above, we do not observe any of the impaired growth, cardiac failure, or postnatal fatality described in [6] demonstrating that their results arise from reduced NRP1 levels in the *Nrp1*^*Y297A/Y297A*^ hypomorph rather than a lack of VEGF-NRP1 binding.

*4) Some mechanistic data related to how NPN1 regulates VEGFR2 level will significantly elevate this manuscript to a different level. The authors should consider including one or two in vitro analysis similar to the NCAM / FGFR1 study (*[10]*)*.

As described in detail above, previous *in vitro* work in multiple cell culture systems has used gain of function (overexpression) and loss of function (RNAi knock-down) to demonstrate that NRP1 is essential for the proper presentation, recycling and degradation of VEGFR2. These *in vitro* studies are consistent with our *in vivo* results and provide clear mechanistic data related to how NRP1 regulates VEGFR2. Our current study provides the first *in vivo* evidence that NRP1 controls VEGFR2 levels at the cell membrane and offers the first *in vivo* phenotypic characterization linking NRP1 regulated VEGFR2 surface expression to vascular development.

Reviewer #2:

*1) The authors are not giving a fair representation of the literature. In the Abstract they report: “Npn1 is essential for vascular morphogenesis, but how Npn1 functions to guide vascular development remains elusive”. This statement is unfair considering several previous papers: Lanahan et al Dev Cell 2013; Fantin et al Development 2014 or Raimondi et al J Exp Med 2014*.

*Furthermore, similarly to what is reported here,*
[6]
*already generated mice expressing Npn1 with a mutation that prevents VEGF binding. Homozygous mutated mice were viable and no severe and lethal cardiovascular phenotypes were observed implying that Npn1 could have VEGF-independent role during vascular development. Although the experimental approaches used in the two papers are different, the final conclusion remains the same*.

We have updated this sentence in the Abstract to provide a fairer representation of the previously published work. As Reviewer 1 points out, the phenotypes described in [6] have two potential interpretations: they could originate from the severe reduction in NRP1 levels or from the abolishment of VEGF-NRP1 binding. Our *Nrp1*^*VEGF-*^ mutant maintains appropriate levels of NRP1 protein (Figure 3, Figure 3–figure supplement 3D) and provides the proper *in vivo* tool to delineate between these two alternatives. We have now assessed the phenotypes described in [6] and while some are linked to VEGF-NRP1 binding (retinal vasculature extension and arteriogenesis) others most likely result from the reduced NRP1 levels observed in the *Nrp1*^*Y297A/Y297A*^ hypomorph rather than from lack of VEGF-NRP1 binding (cardiac failure, retinal vascular density, and retinal endothelial tufts). A detailed point-by-point description of these results is provided above.

*2) The reduction in VEGFR2 levels in Npn1 mutants is indeed striking but does not correspond to the findings of many other authors where the uncoupling or strong reduction on Npn1 did not modify VEGFR2 or VEGFR1 expression (see, as an example, Fantin et al. Development 2014). In addition, while in*
Figure 5
*the drop in receptor expression is almost complete; in*
Figure 5
*it is around 35% only*.

*Since Title, Abstract and Discussion are focused on this finding the authors should present more solid data to support their conclusions. Last but not least, if the decrease in VEGFR2 is in the range presented in*
Figure 5
*the phenotype of Npn1 KO should be comparable to that of VEGFR2 KO and this is not the case*.

We agree with this sentiment and have gone back to quantify the VEGFR2 levels in our Western blots. Indeed, the decrease in VEGFR2 levels is more on par with the FACS analysis and we have replaced the Western blot image with a more representative data. Furthermore, we have included our quantification in Figure 5.

*3) The authors do not investigate the possible effects of their Npn1 mutant on neurons. Taking into account that some types of neurons rely on VEGF165 and Npn1 signaling, these observations are needed for a more complete interpretation of the mutant phenotype*.

First, we looked at the most robust neural phenotype that has been linked to VEGF-NRP1 function. Specifically, facial motor neurons express NRP1, and their migration during development is dependent on the presence of VEGF164 (Schwarz et al., 2004). According to the literature, *Nrp1*^*-/-*^ mutants have a defect in which the migration of the facial motor neurons is disrupted. The phenotype was thought to be a result of VEGF-NRP1 binding in neurons because *Nrp1*^*Sema-*^ and *Tie2-Cre;Nrp1*^*fl/-*^ mutants do not have this phenotype while VEGF-120 mice phenocopy the *Nrp1*^*-/-*^. With *in situ* hybridization for *Isl1*, we analyzed the migratory path of motor neurons at E12.5. To our surprise, we did not see a migration defect in the *Nrp1*^*VEGF-*^ mutant. However, we also failed to see a migration defect in the neuronal-specific NRP1 knock-out (*Nestin-Cre,Nrp1*^*fl/fl*^) demonstrating that NRP1 is not required in neurons for this phenotype. In this regard, our data bring the claims of previous work into question and resolving this discrepancy is beyond the scope of our current manuscript. In addition, we also examined gonadotropin-releasing hormone (GnRH) neurons because Carboni et al., 2011 showed that VEGF164 promotes the survival of migrating GnRH neurons by co-activating the ERK and AKT signaling pathways through NRP1. However, as with the facial motor neurons, we did not detect any GnRH survival phenotype in our *Nrp1*^*VEGF-*^ mutants.

*4) Introduction. Most of previously published data converge in saying that the domain of Npn1 responsible for the binding of Sema 3a is the b1 region and not the a1 (see for instance*
Figure 2
*Nat. Rev. Immunol. 2103 by Kikutani). This should be introduced and discussed in a better way in the text*.

We have already stated that “A previous structure-function analysis revealed that the b1 domain of NRP1 is necessary and sufficient for VEGF binding (12). However, this b1 region is also required for SEMA3-NRP1 interactions so a series of *Nrp1* variants containing smaller deletions in the b1 domain were engineered with site-directed mutagenesis to identify a region specific for VEGF-NRP1 binding (Figure 1)” in the Results section.

Reviewer #3:

*Furthermore, the authors found that the Nrp1D320K mutation caused a dramatic reduction of VEGFR expression. This is a very intriguing observation and it would be interesting to know the reason for this and what the functional consequences are. The study does not explore this but the authors conclude (in the Discussion) that “Nrp1 regulates angiogenesis by controlling the amount of VEGFR2 expression at the cell surface and consequently the level of VEGFR2-VEGF signaling*.*”*

*There are several problems with this statement*:

*1) It is logically wrong. Since the Npn1VEGF mice have reduced VEGFR2 levels but develop normally, Nrp1 obviously is not regulating angiogenesis via VEGFR2 levels, otherwise the mice would have a Nrp1 KO phenotype*.

Based on the critique, it is apparent that Reviewer 3 misunderstood the results from our study. We clearly show that *Nrp1*^*VEGF-*^ mice have normal VEGFR2 levels, as seen in Figure 5 via Western blot and FACS analysis. In contrast, *Tie2-Cre,Nrp1*^*fl/-*^ mutants have reduced VEGFR2 levels which is consistent with the *Nrp1*^*-/-*^ phenotype shown in Figure 4.

*2) The authors have not actually shown that VEGFR2 signalling is reduced. This would have to be shown more directly (e.g. VEGFR2 phosphorylation in Western blots)*.

We have performed additional experiments to determine the level of VEGFR2 phosphorylation in lung lysates following VEGF treatment in the *Tie2-Cre,Nrp1*^*fl/-*^ and *Nrp1*^*VEGF-*^ mutants. Please see our detailed description above.

*3) Although the flow cytometry data in*
Figure 5
*indicates reduced surface expression of VEGFR2 this seems only to be the case for around 50% of the cells*. *On the other hand, the Western blot data (*Figure 5*) indicates a much stronger reduction of around 90% (contradicting the flow cytometry data somewhat). This demonstrates that also intracellular VEGFR2 must be dramatically reduced and not only surface VEGFR2*

As previously described above, we have reassessed VEGFR2 levels in our Western blots and now have included quantification in Figure 5. Importantly, we have determined that the decrease in VEGFR2 observed in the *Tie2-Cre,Nrp1*^*fl/-*^ mutant is comparable to the reduction seen in our FACS analysis

*4) VEGFR2 levels were only studied in lung endothelial cells. Yet the main vascular phenotype in Nrp1 mutants is found in the brain and retinal vasculature*.

We examined VEGFR2 levels in lung endothelial cells for practical purposes, in that we were able to harvest the greatest number of endothelial cells from this organ to perform our analysis. The *Tie2-Cre;Nrp1*^*fl/-*^ mutants are embryonic lethal and finding a tissue with an adequate source of cells was of paramount importance. We attempted to analyze brain endothelial cells via FACS, but the tissue was difficult to dissociate and formed clumps during the immunolabeling process. These clumps were unable to be analyzed via the FACS equipment and resulted in a significant loss of endothelial cells from our samples.

*5) The Nrp1Y297A mice also developed normally but that study found abnormalities in postnatal animal, such as abnormal retinal vasculature development, reduced neovascularisation in the oxygen induced retinopathy model and reduced tumour growth. These postnatal phenotypes should be also checked in the Nrp1D320K mice*.

We appreciate this constructive criticism and have now included a broader phenotypic analysis of the *Nrp1*^*VEGF-*^ mutant. In particular, we have examined the retinal vasculature at two time points (P9 and adult), challenged the mutant with a hindlimb ischemia assay, and included additional information about fertility, organ size, and viability.